# From Boltzmann Machines to Neural Networks and Back Again

**Surbhi Goel**
Microsoft Research NYC
surbgoel@microsoft.com

**Adam Klivans**
University of Texas at Austin
klivans@cs.utexas.edu

**Frederic Koehler**
MIT
fkoehler@mit.edu

## Abstract

Graphical models are powerful tools for modeling high-dimensional data, but learning graphical models in the presence of latent variables is well-known to be difficult. In this work we give new results for learning Restricted Boltzmann Machines, probably the most well-studied class of latent variable models. Our results are based on new connections to learning two-layer neural networks under $\ell_\infty$ bounded input; for both problems, we give nearly optimal results under the conjectured hardness of sparse parity with noise. Using the connection between RBMs and feedforward networks, we also initiate the theoretical study of *supervised RBMs* [1], a version of neural-network learning that couples distributional assumptions induced from the underlying graphical model with the architecture of the unknown function class. We then give an algorithm for learning a natural class of supervised RBMs with better runtime than what is possible for its related class of networks without distributional assumptions.

## 1 Introduction

Graphical models are a powerful framework for modelling high-dimensional distributions in a way that is interpretable and enables sophisticated forms of inference and reasoning. They are extensively used in a variety of disciplines including the natural and social sciences where they have been used to model the structure of gene regulatory networks, of connectivity in the brain, and the flocking behavior of birds [2]. In many contexts, the structure of interactions between different observed variables is unknown a priori and the goal is to infer this structure in a sample-efficient way from data. There has been decades of research on various formulations of this problem, both theoretically and empirically: for example, provable algorithms have been developed for learning tree-structured graphical models [3], for learning models on graphs of bounded tree-width [4], for learning Ising models on general graphs of bounded degree [5, 6, 7, 8] and in a variety of other contexts like Gaussian graphical models (e.g. [9]). For the most part, the main interest has been on learning under the assumption that the underlying model is sparse. Sparsity is a natural assumption since many applications are in a sample-starved regime where the learning problem is information-theoretically impossible without sparsity. Sparse models are generally considered to be more interpretable than their dense counterparts since they satisfy large numbers of conditional independence relations.

A major challenge in probabilistic inference from data is the presence of latent or confounding variables which are unobserved and may create complicated higher-order dependencies between the observed variables. Specifically in the context of learning undirected graphical models, it is well known that even if the underlying graphical model is well-behaved, if only a subset of the variables are observed then the resulting marginal distribution can still be extremely complicated, e.g. simulating the uniform distribution over satisfying assignments of an arbitrary circuit [10], which makes the learning problem computationally intractable. On the other hand, under certain assumptions we know that learning graphical models with latent variables can be both computationally

and statistical tractable; for example, the setting of tree-structured models with latent variables has been extensively studied in the context of phylogenetic reconstruction, see e.g. [11, 12]. However, in non-tree-structured models there are comparatively few positive results for recovering latent variable models in a computationally efficient fashion. One of the few exceptions is in the Gaussian case, where [13] gave a positive result; this setting is very special, as latent variable GGMs *do not* have higher-order interactions, but in fact are equivalent to GGMs with cliques.

**Restricted Boltzmann Machines.** In this work, we will focus on a latent variable model popularized in the neural network literature known as the *Restricted Boltzmann Machine* (RBM) (see e.g. [1, 14]) which has been applied to problems such as dimensionality reduction and collaborative filtering [15, 16, 17]. It is also perhaps the most canonical version of an Ising model with latent variables. The RBM describes a joint distribution over observed random variables $X$ valued in $\{\pm 1\}^{n_1}$ and latent variables $H$ valued in $\{\pm 1\}^{n_2}$

$$\Pr(X = x, H = h) \propto \exp\left(\langle x, Wh \rangle + \langle b^{(1)}, x \rangle + \langle b^{(2)}, h \rangle\right)$$

where the *weight matrix* $W$ is an arbitrary $n_1 \times n_2$ matrix and *external fields/biases* $b^{(1)} \in \mathbb{R}^{n_1}$ and $b^{(2)} \in \mathbb{R}^{n_2}$ are arbitrary, and $X$ is referred to as the vector of *visible unit* activations and $H$ the vector of *hidden unit* activations. In the learning problem, we are given access to i.i.d. samples of $X$ but do not get to observe $H$. It is not hard to see that in the special case where the hidden nodes are constrained to have degree 2, the class of marginal distributions on $X$ induced by RBMs is exactly the class of Ising models (pairwise binary graphical models), so the general RBM can be thought of as a natural generalization of fully-observed Ising models, for which the learning problem is well-understood. For hidden units with larger degree, the marginal distribution can be an arbitrary Markov Random Field [18]. We also note that the parameters of the RBM are not identifiable even given an infinite number of samples [18], so our goal for learning the RBM is generally speaking to learn the distribution or related structural properties (e.g. the Markov blankets of the nodes in $X$).

**Previous work on Learning RBMs.** The most popular heuristic for learning RBMs is the *contrastive divergence* algorithm (see [1]), but there is no guarantee it will succeed. In recent work [18, 19], the first provable algorithms were developed for learning RBMs, under the assumptions that the model is (1) sparse and (2) ferromagnetic. On the other hand, it was shown in [18] that learning general sparse RBMs is computationally intractable in general, because the conjecturally hard problem of learning a *sparse parity with noise* [20] can be embedded into a sparse RBM with a constant number of hidden units. The assumption of ferromagneticity (that variables are only positively correlated, not negatively correlated) rules out this example and plays a crucial role in the analysis of these works. Without ferromagneticity, viewing the marginal on $X$ as a general Markov Random Field allows for using prior work [8] to give learning algorithms with runtime $n^{O(d_H)}$ where $d_H$ is the maximum degree of a hidden node. This matches the lower bound of learning sparse parity with noise mentioned previously.

To summarize, the best previous results for learning RBMs either (1) make the assumption of ferromagneticity which makes building sparse parities impossible or (2) ignore all of the structure of the RBM except the max hidden degree, and pay the price of a $n^{\Theta(d_H)}$ runtime. This leaves open the question of developing algorithms whose runtime depends on some natural notion of a *complexity* measures of the RBM.

**Our Results.** In this paper, we design an algorithm that is adaptive to a *norm* based complexity measure of the RBM, and often outperforms approach (2) above significantly, while not eliminating the possibility of negative correlation completely as in (1). The key idea of our approach is to develop a novel connection between learning RBMs and their historical relative, feedforward neural networks. This connection allows us to establish new results for learning RBMs, by proving new results about learning feedforward neural networks (Section 2).

Our connection also validates the idea of a so-called *supervised RBMs* as a natural distributional setting for classification with feedforward networks. Supervised RBMs, proposed by Hinton [1], treat one visible unit of the RBM as the label and the other visible units as the input to the classifier. This allows us to use the connection in the "reverse" direction — using natural structural assumptions on the RBM (like ferromagneticity) to give better results for solving supervised prediction tasks in an interesting distributional setting. Along these lines, we show that an assumption related to

ferromagneticity, but allowing for some amount of negative correlation in the RBM, allows us to learn the induced feedforward network faster than would be possible without distributional assumptions (Section 3). Lastly, we present an experimental evaluation of our "supervised RBM" algorithm on MNIST and FashionMNIST to highlight the applicability of our techniques in practice (Section 5).

## 2 Learning RBMs via New Results for Feedforward Networks

**Relationship between RBMs and Feedforward Networks**    Our first result characterizes the relationship between RBMs and Feedforward networks. We show that there is a natural self-supervised prediction task in RBMs, of predicting the spin at node $i$ given all other observed nodes, for which the Bayes-optimal predictor is *exactly given* by a two-layer feedforward network with a special family of $\tanh$-like activations.

**Theorem 1.** *For any visible unit $i$ in an arbitrary RBM,*

$$\mathbb{E}[X_i|X_{\sim i}] = \tanh\left(b_i^{(1)} + \sum_j \tanh(W_{ij})f_{\beta_{ij}}\left(b_j^{(2)} + \sum_{k \neq i} W_{kj}X_k\right)\right) \tag{1}$$

*where $\beta_{ij} = |\tanh(W_{ij})|$ and $f_\beta(x) := \frac{1}{\beta}\tanh^{-1}(\beta\tanh(x))$.*

*Proof.* Observe that the conditional distribution of $(X_i, H)$ given $X_{\sim i} = x_{\sim i}$ is given by

$$\Pr(X_i = x_i, H = h|X_{\sim i} = x_{\sim i}) \propto \exp\left(x_i(b_i^{(1)} + \sum_j W_{ij}h_j) + \langle W_{\sim i}^t x_{\sim i} + b^{(2)}, h\rangle\right) \tag{2}$$

where $W_{\sim i}$ denotes the $(n_1 - 1) \times n_2$ dimensional matrix given by deleting row $i$. Since the only quadratic terms left in the potential are between the remaining visible unit $X_i$ and the hidden units $h_j$, this conditional distribution is exactly an Ising model on a star graph, i.e. a tree of depth 1 with root node corresponding to $X_i$. For all tree-structured graphical models, the conditional distribution of the root given the leaves can be computed exactly by Belief Propagation (see e.g. [21, 22]); in the case of Ising models it's known the general BP formula can be written with hyperbolic functions as above[1]. □

**Remark 1.** *An analogous result can be proved in the more general setting where the spins do not have to be binary; for example in a Potts model version of the RBM where each spin is valued in a set of size q, the conditional law of $X_i$ given the others would be given again by a two-layer network where the last layer is a softmax. In this paper we focus on the binary case for simplicity.*

**Remark 2.** *The family of activation functions $f_\beta(x)$ naturally interpolates between the identity activation ($\beta = 1$ where $f_\beta(x) = x$) and $\tanh$ activation at $\beta = 0$, since*

$$\lim_{\beta \to 0} \frac{1}{\beta}\tanh^{-1}(\beta\tanh(x)) = \frac{\partial}{\partial\beta}\tanh^{-1}(\beta\tanh(x))\Big|_{\beta=0} = \tanh(x).$$

The exact structure of this prediction function is crucial in what follows and does not seem to have been known in the RBM literature, though some related ideas have been used to develop better heuristics for performing inference and training in RBMs (see discussion in Appendix B).

Given this connection, we show that if we can solve the problem of learning such a neural network within sufficiently small error, then we can successfully learn the RBM. This reduces our RBM learning problem to that of learning feedforward neural networks in the setting that the input is bounded in $\ell_\infty$ norm.

**Improved Results for Learning Feedforward Networks**    Subsequently, we give results for the feedforward network problem which are nearly optimal both in the terms of sample complexity (in the regime where $\lambda$ is bounded) and in terms of computational complexity under the hardness of learning sparse parity with noise; some aspects of this result are new even for the well-studied case of learning neural networks with $\tanh$ activations (see Further Discussion).

**Theorem 2** (Informal version of Corollary 1). *Suppose that $Y$ is a random variable valued in $\{\pm 1\}$, $X$ is a random vector such that $\|X\|_\infty \leq 1$ almost surely and*

$$\mathbb{E}[Y|X] = \tanh\left(b^{(1)} + \sum_j w_j f_{\beta_j}\left(b_j^{(2)} + \sum_k W_{jk} X_k\right)\right)$$

*where $b^{(1)} \in \mathbb{R}$, $\beta_j \in [0,1]$, $w$ is an arbitrary real vector and $W$ is an arbitrary real matrix. Let $W_j$ denote column $j$ of $W$ and suppose $\|W_j\|_1 \leq \lambda$ for every $j$ and some $\lambda \geq 2$. Then if we run $\ell_1$-constrained regression on the degree $D$ monomial feature map $\varphi_D(x) \mapsto \left(\prod_{i \in S} X_i\right)_{|S| \leq D}$ with appropriate $\ell_1$ constraint, the result $\hat{w}$ satisfies with high probability*

$$\mathbb{E}[\ell(\hat{w} \cdot \varphi_d(X), Y)] \leq OPT + \epsilon$$

*where $OPT$ is the minimum logistic loss for any measurable function of $X$, as long as the number of samples $m$ satisfies $m = \Omega((|b^{(1)}|^2 \lambda^{O(D)}) \log(2n))$ where $D = O(\lambda \log(\|w\|_1 \lambda/\epsilon))$ and the runtime of the algorithm is $poly(n^D)$.*

We also show, under the standard assumption for hardness of learning sparse parity with noise, the following lower bound which shows that the runtime guarantee in our result is close to tight even in the usual setting of $\tanh$ neural networks ($\beta_j = 0$) — it is optimal up to $\log \log$ factors in the exponent in its dependence on $\epsilon$ and $\|w\|_1$, and we also show that at least a subexponential dependence (essentially $2^{\sqrt{\lambda}}$) on $\lambda$ is unavoidable (assuming the dependence on other parameters in the statement is fixed, since there are e.g. trivial algorithms that run in time $2^n$).

**Theorem 3** (Informal version of Theorem 11). *There exists families of models (one with $\epsilon$ a constant, one with $\|w\|_1$ a constant) where a runtime of $n^{\Omega\left(\frac{\log(\|w\|_1/\epsilon)}{\log \log(\|w\|_1/\epsilon)}\right)}$ is needed for any algorithm to achieve $\epsilon$ error with high probability, regardless of its sample complexity. Even in the case of $\tanh$ activations ($\beta_j = 0$ for all $j$), there exists a sequence of models with $\lambda = \Theta(n \log(n))$ and $\|w\|_1 = O(\sqrt{n})$ which requires runtime $n^{\Omega(\sqrt{\lambda/\log^2(\lambda)} \log(n) \log \|w\|_1)}$ to achieve error $\epsilon = 0.01$ with high probability.*

To our knowledge, the fact that $n^{\log(\|w\|_1/\epsilon)/\log\log(\|w\|_1/\epsilon)}$ runtime is required to learn this class even for $\lambda = 1$, and by the above upper bound is tight up to the $\log \log$ term, was not known before even for standard $\tanh$ networks. As far as the dependence on $\lambda$, a similar problem was studied in [23] where they proved the dependence cannot be polynomial using the result of [24] for intersection of halfspaces, based on a different assumption, though our lower bound seems to be somewhat stronger in the present context.

In particular the lower bounds on the runtime show that methods like the kernel trick cannot significantly improve the runtime compared to the simple method of writing out the feature map explicitly used in Theorem 2; however, writing out the feature map lets us use $\ell_1$ regularization[2] instead of $\ell_2$ which can give significant sample complexity advantages (e.g. $O(\log n)$ vs $O(n)$ for the usual sparse linear regression setups).

**Structure Learning of RBMs**   As explained above, our reduction based on Theorem 1 lets us use the above feedforward network learning result to learn the structure of RBMs. By structure learning, we mean learning the *Markov blanket* of the each visible unit in the marginal distribution of the RBM over visible units, i.e. the minimal set of nodes $S$ such that $X_i$ is conditionally independent of all other $X_j$ conditionally on $X_S$. We will also refer to the Markov blanket as the (two-hop) neighborhood of node $i$. This is a natural objective as other tasks such as distribution learning are straightforward in sparse models if the Markov blankets are known. As in the previous work on structure learning in other undirected graphical models (e., we will need some kind of quantitative nondegeneracy condition to guarantee nodes in the Markov blanket of node $i$ are information-theoretically discoverable; it is not hard to see (e.g. using the bounds from [26]) that if two nodes are neighbors but their interaction is extremely weak then it becomes impossible to distinguish the model from the same model with the edge removed without a very large number of samples.

In Ising models and in ferromagnetic RBMs, there are simple conditions on the weight matrices which can ensure neighbors are information-theoretically discoverable. In a general RBM, there is no natural way to place constraints on the weights of the RBM to ensure this: the issue is that two nodes $X_i$ and $X_j$ can be independent even though they have two neighboring hidden units with non-negligible edge weights, since the effect of those hidden units can exactly cancel out so that $X_i$ and $X_j$ are independent or indistinguishably close to independent (a number of examples are given in [18]). For this reason, we will instead make the following assumption on the behavior of the model itself instead of on its weight matrix:

**Definition 1.** *We say that visible nodes $i, j$ are $\eta$-nondegenerate two-hop neighbors if*

$$I(X_i; X_j | X_{\sim i,j}) = \mathbb{E}[\ell(\mathbb{E}[X_i | X_{\sim\{i,j\}}], X_i)] - \mathbb{E}[\ell(\mathbb{E}[X_i | X_{\sim i}], X_i)] \geq \eta$$

*or if the same inequality holds with $i$ and $j$ interchanged. Here $I(X_i; X_j | X_{\sim i,j})$ is the conditional mutual information between $X_i$ and $X_j$ conditional on $X_{\sim i,j}$, and the equality follows from Fact 1 in the Appendix and the definition of mutual information in terms of KL [27].*

Information-theoretically, this condition says that nontrivial information is gained about $X_i$ by observing $X_j$, even after we have already observed $X_{\sim i,j}$. The fact that $X_j$ is in the Markov blanket of node $X_i$ exactly means that this quantity is nonzero. By Pinsker's inequality [27], $\eta$-nondegeneracy is also implied by a lower bound on the partial correlation $\text{Cov}(X_i, X_j | X_{\sim i,j})$.

**Example 1.** *It is not hard to see that Ising models are equivalent to the marginal distribution of RBMs with maximum hidden node degree equal to $2$. Consider an Ising model with minimum edge weight $\alpha$ and such that the maximum $\ell_1$-norm into every node is upper bounded by $\lambda$ and the external field is upper bounded by $B$, then $\eta \geq e^{-O(\lambda+B)}/\alpha$, see e.g. [6].*

**Example 2.** *In a ferromagnetic RBM with minimum edge weight $\alpha$ and maximum external field $B$, it can be shown that $\eta \geq e^{-O(\lambda_1+\lambda_2+B)}/\alpha^2$ (see [18, 19]).*

In order for the RBM to be learnable with a reasonable number of samples (since general RBMs can represent arbitrary distributions with full support on the hypercube [18]), we need to assume it has low complexity in the following sense:

**Definition 2.** *We say that an RBM is $(\lambda_1, \lambda_2)$-bounded if for any $i$, $\sum_j |\tanh(W_{ij})| + |b_i^{(1)}| \leq \lambda_1$ and the columns of $W$ are bounded in $\ell_1$ norm by $\lambda_2$.*

Note that $\lambda_1$ and $\lambda_2$ bound the $\ell_1$ norm into the visible and hidden units, respectively. Based on our upper bounds and lower bounds for the learnability of feedforward networks, it should be less surprising that these parameters play a very different role in the computational learnability of RBMs.

**Theorem 4** (Informal version of Theorem 12). *Suppose all two-neighbors in a $(\lambda_1, \lambda_2)$-bounded RBM are $\eta$-nondegenerate. Given $m = \Omega(\lambda_2^{O(D)} \log(2n))$ i.i.d. samples from the RBM, where $D = O(\lambda_2 \log(\lambda_1\lambda_2/\eta))$, we can recover its structure with high probability in time $poly(n^D)$.*

Based on this result we also give a result for learning the RBM in TV distance under the same assumption: see Theorem 13: the sample complexity of this method is essentially the above sample complexity plus $n^2(1 - \tanh(\lambda_1))^{-d_2}$ where $d_2$ is the maximum 2-hop degree; the $poly(n)$ dependence is required as even learning $n$ bernoullis in TV requires $\Omega(n)$ sample complexity. Our algorithm encodes the distribution as a sparse Markov Random Field, but (if desired) this can easily be converted into a sparse RBM using an algorithm in [18]. Therefore we learn the distribution properly, except that the learned RBM typically has more hidden units than the original RBM (i.e. it is overparameterized).

When interpreting these result, it is crucial not to confuse the $\ell_1$ norm parameters $\lambda_1, \lambda_2$ of visible and hidden units with the maximum degrees of these units. Typically in Ising models, we should think of the weight of a typical edge as *shrinking* as $d$ grows so that units stay near the sensitive region of their activation and the behavior of the model does not become trivial — this means that $\lambda_1$ and $\lambda_2$ may be much smaller than $d$. This is consistent with practical advice in the RBM literature, see e.g. [1]. Probably the most well known sufficient condition for being able to sample in an Ising model (or RBM) is *Dobrushin's uniqueness criterion* which is equivalent to the requirement that $\lambda_1, \lambda_2 \leq 1$ and this condition is actually tight for Glauber dynamics to mix quickly in the Ising model on the complete graph (Curie-Weiss Model) [28]. We discuss this further in Remark 5; in Dobrushin's uniqueness regime and under some mild nondegeneracy conditions we expect that $\eta = \Omega(1/d^2)$

so the above algorithm has runtime $n^{\log(d)}$, which is an exponential improvement in the exponent compared to the best previously known result ($O(n^d)$ runtime by viewing the RBM as an MRF).

We also give lower bound results showing that the computational complexity of the above algorithm is essentially optimal in terms of $\lambda_1$ and $\eta$ (based upon the hardness of learning sparse parity with noise) and nearly optimal in terms of $\lambda_2$ for an SQ (Statistical Query) algorithm, in the sense that any SQ algorithm needs at least sub-exponential dependence on $\lambda_2$ (given that the dependence on other parameters is not changed — e.g. obviously there is a $2^n$ time algorithm to learn this problem). In particular, this shows that our results for learning feedforward networks under $\ell_\infty$ are close to tight even in this application, where the input distribution is related to the label.

**Theorem 5** (Informal version of Theorem 19). *As before, $\lambda_2$ refers to the maximum $\ell_1$-norm into any hidden unit and we choose parameters so that $\lambda_2 = poly(n)$ and $\lambda_1 = poly(n)$. There exists $\epsilon > 0$ so that no SQ algorithm with tolerance $n^{-\lambda_2^\epsilon}$ and access to $n^{\lambda_2^\epsilon}$ queries can structure learn an $\alpha = \Omega(1)$-nondegenerate $(\lambda_1, \lambda_2)$-bounded RBM.*

We also show (Theorem 16) that the $\eta$-nondegeneracy condition is required to achieve nontrivial guarantees even if we are only interested in distribution learning (i.e. in TV), assuming the hardness of learning sparse parity with noise.

## 3 Supervised RBMs

Since in many applications the input data to a classifier is clearly very structured (e.g. images, natural language corpuses, data on networks, etc.), it is interesting to consider the behavior of classification algorithms under structural assumptions on the data. RBMs are one (relatively simple) generative model which can generate interesting structured data. This suggests the idea of learning "supervised RBMs", as proposed by Hinton [1], where we assume the input and label are drawn from an RBM joint distribution, so that predicting the label is a feedforward network by Theorem 1; in this model the label is just a special visible unit in the RBM. Based on the previous discussion about computational lower bounds, we know that assuming the input to a feedforward network comes from the corresponding RBM does not in general make learning easier, but we know that in RBMs there are very natural assumptions we can make to avoid these computational issues. Our final result is of exactly this flavor, showing how we can learn the supervised RBM under a ferromagneticity-related condition faster than is possible if we did not have a distributional assumption.

In order to emphasize the special role of the node which we want to predict, we will adopt a modified notation where the visible unit which we want to learn to predict is labeled $Y$ and all other visible units are still labeled $X$. More precisely, we model the joint distribution over input features $X$ valued in $\{\pm 1\}^{n_1}$, latent features $H$ valued in $\{\pm 1\}^{n_2}$ and label $Y \in \{\pm 1\}$ as,

$$\Pr[X = x, H = h, Y = y] \propto \exp\left(\langle x, Wh \rangle + \langle h, w \rangle y + \langle b^{(1)}, x \rangle + \langle b^{(2)}, h \rangle + b^{(3)} y\right)$$

where the *weight matrix* $W$ is a non-negative $n_1 \times n_2$ matrix, $w$ is an arbitrary $n_1$ dimensional vector and $b^{(1)} \in \mathbb{R}^{n_2}, b^{(2)} \in \mathbb{R}^{n_2}$ and $b^{(3)} \in \mathbb{R}$ are arbitrary. Given the latent variables $H$, $w$ can be seen as the linear predictor for $Y$.

**Theorem 6** (Informal Version of Theorem 21). *Suppose the interaction matrix $W$ is ferromagnetic with minimum edge weight $\alpha$. Further suppose one of the RBMs induced by conditioning on $Y = 1$ or $Y = -1$ is a $(\lambda, \lambda)$-RBM. Then there exists an algorithm that learns the predictor $Y$ that minimizes logistic loss up to error $\epsilon$. The algorithm has sample complexity $m = n_1^2 \exp(\lambda)^{\exp(O(\lambda))} (1/\alpha)^{O(1)} \log(n_1/\delta)/\epsilon^2$ and has runtime $poly(m)$.*

Our main algorithm can be broken down into three main steps: (1) Use greedy maximization of conditional covariance $\mathsf{Cov}^{\mathsf{Avg}}$ to first learn the two-hop neighborhood $\mathcal{N}(i)$ of each observed variable $i$ w.r.t. the hidden layer conditioned on the label (see Algorithm 1), (2) For each observed variable $X_i$, learn the conditional law of $X_i \mid X_{\mathcal{N}(i)}, Y$ using regression, and (3) Use the estimated distribution to compute $\mathbb{E}[Y|X]$. Step (1) leverages tools from [18, 19] but considers a setting where the RBM may in fact have some amount of negative correlation, as $w$ has arbitrary signs and is allowed to have large norm. Step (2) can be achieved by simply looking at the conditional law under the empirical distribution; this is efficient as we learn small neighborhoods.

In step (3), we can make use of the following useful trick (a version of which can be found in [1]): we already have enough information to derive the law of $Y \mid X$ since we know the marginal law of $Y$ (the fraction of $+$ and $-$ labels) and the law of $X \mid Y$. However, naively carrying out the Bayes law calculation is difficult because it involves partition functions (which are in general NP-hard to approximate, see e.g. [29]). We avoid computing the partition function by observing that if we define $f_1, f_2$ such that $\Pr(X, Y) \propto \exp(f_1(X)\mathbb{1}(Y = 1) + f_2(X)\mathbb{1}(Y = -1) + by)$, then the law of $Y \mid X$ follows a logistic regression model where

$$\mathbb{E}[Y \mid X] = \tanh\left(\frac{f_1(X) - f_2(X)}{2} + b\right)$$

for some constant $b \in \mathbb{R}$. Therefore if we know $f_1, f_2$ up to additive constants (which we can derive from the Fourier coefficients learned in (2)), we can simply fit a logistic regression model from data to learn $h$ plus the missing constants, and we can prove this works using fundamental tools from generalization theory. We refer the reader to Appendix E for additional details.

---

**Algorithm 1** LEARNSUPERVISEDRBMNBHD$(u, \tau, \mathcal{S})$ (Adapted from [18, 19])

---

1: Set $S := \phi$
2: Set $i^* = \arg\max_v \widehat{\mathsf{Cov}}_{\mathcal{S}}^{\mathsf{Avg}}(u, v|S, Y)$, and $\eta^* = \max_v \widehat{\mathsf{Cov}}_{\mathcal{S}}^{\mathsf{Avg}}(u, v|S, Y)$
3: **if** $\eta^* \geq \tau$ **then**
4:     $S = S \cup \{i^*\}$
5: **else**
6:     Go to Step 8
7: Go to Step 2
8: For each $v \in S$, if $\widehat{\mathsf{Cov}}_{\mathcal{S}}^{\mathsf{Avg}}(u, v|S\backslash\{v\}, Y) < \tau$, remove $v$ (*Pruning step*)
9: Return $S$

---

Observe that under the given distributional assumptions, our algorithm has runtime complexity polynomial in the input dimension in contrast to Theorem 2 where the run time scales as $n^{\tilde{\Omega}(\lambda)}$. A simple example which shows the algorithm from this Theorem will outperform any algorithm without distributional assumptions (like Theorem 2) is given in Remark 8.

## 4   Discussion: Comparison to Prior work on Learning Neural Networks

In the neural network learning literature, various works prove positive results that either (1) work for any distribution with norm assumptions or (2) require strong distributional assumptions. The result of Theorem 2 falls into the category (1) and the result of Theorem 6 falls into category (2).

We first discuss the relation of Theorem 2 to other previous works of type (1). Perhaps the most closely related works are [23, 30, 31, 32]. All of these works assume the input is bounded in $\ell_2$ norm and give learning results based on kernel methods; of course, these results could be applied under the assumption of $\ell_\infty$-bounded input, by using the inequality $\|x\|_2 \leq \sqrt{n}\|x\|_\infty$ and rescaling the input to have norm 1. For comparison, the best result in the $\ell_2$ setting with $\tanh$ activation is given in [32], but this result (as is essentially necessary based on the known computational hardness results) has exponential dependence on the $\ell_2$ norm of the weights in the hidden units, so doing such a reduction just using norm comparison bounds gives a runtime sub-exponential in dimension. Therefore it is indeed crucial for us to give a new analysis adapting to learning with input bounded in $\ell_\infty$. An interesting feature of this setting (as mentioned above) is that the kernel trick does not seem to be as useful for improving the runtime as the $\ell_2$ setting, where it seems genuinely better than writing out the feature map [31, 32].

Due to the generality of direction (1), it is hard to design efficient algorithms. This further motivates direction (2), however, making the right distributional assumptions which allow for efficient learning while being well-motivated in context of real world data can be very challenging. Most prior work has been limited to the Gaussian input [33, 34, 35, 36, 37, 38] or symmetric input [32, 39] assumptions which are not satisfied by real world data. The works of [40, 41] gave results for some simple tree-structured generative models. There has been some work in defining data based notions such as eigenvalue decay [42] and score function computability [43] to get efficient results. Our assumption

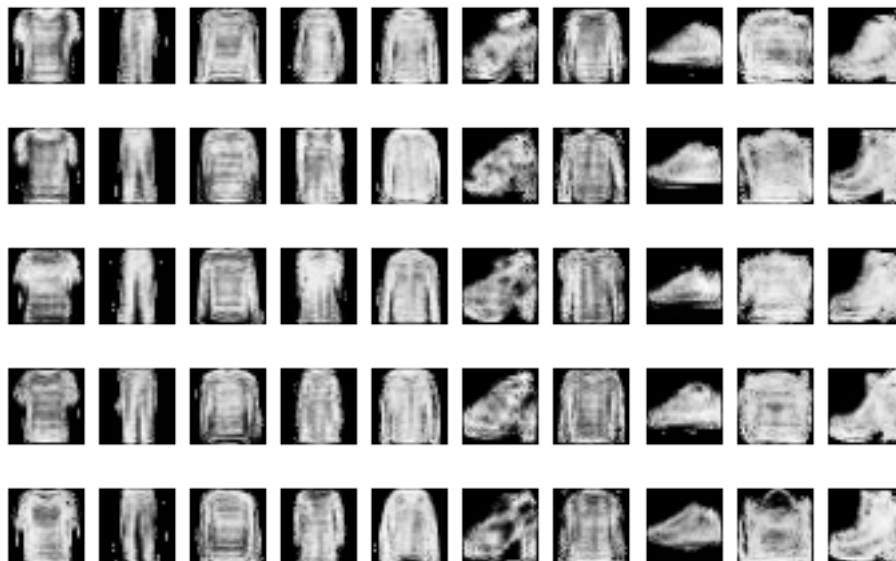

Figure 1: Five i.i.d. samples for each FashionMNIST class, drawn from the trained model by Gibbs sampling.

for Theorem 6 in contrast exploits sparsity and nonnegative correlations among the input features conditional on the output label.

## 5 Experiments

In this section we present some simple experiments on MNIST and FashionMNIST to confirm that our method performs reasonably well in practice. In these experiments, we implemented the supervised RBM learning algorithm from Theorem 6 which makes use of the classification labels provided in the training data set. This algorithm outputs both a classifier (which predicts the label given the image) and also a generative model (which can sample images given a label).

For classification, we allowed the logistic regression (described as "step (3)" above) to fit not just the bias term but also coefficients on the sum of Fourier coefficients for each pixel (an input of dimension $768 \times 10 = 7680$), since the runtime of the logistic regression step is almost negligible anyway. This is useful because it allows greater dynamic range in the influence of each pixel.

We observed a test accuracy of $97.22 \pm 0.16\%$ on MNIST; the training accuracy was $99.9\%$ and we trained the logistic regression for 30 epochs (same as steps) of L-BFGS with line search enabled. For FashionMNIST, we obtained a test accuracy of $88.84 \pm 0.31\%$; the training accuracy was $92.19\%$ and we trained the logistic regression for 45 epochs with L-BFGS as before. Overall training took a bit less than an hour each on a Kaggle notebook with a P100 GPU. Both datasets have $60,000$ training points and $10,000$ test; in both experiments we used a maximum neighborhood size of 12, and stopped adding neighbors if the conditional variance shrunk by less than $1\%$.

For context, we note that our accuracy on MNIST is better than what we would get using standard training methods for RBMs and logistic regression for classification; [44] reports accuracies of approximately $95\%$ for CD and $96\%$ using a more sophisticated TAP-based training method. The results are also around as good or better than what is achieved using many classical machine learning methods on these datasets [45]; for example, logistic regression achieves error $91.7\%$ and $84.2\%$ and polynomial kernel SVM achieves error $89.7\%$ and $97.6\%$ [45]. Of course, none of these results are as good as specialized deep convolutional networks (over $99\%$ on MNIST). In contrast to other approaches using linear models such as kernel SVM, our approach also learns a generative model. Being able to sample from the generative model can give some insight into how the model classifies.

To evaluate the performance of the learned RBM as a generative model, we generated samples using Gibbs sampling starting from random initialization and run for 6000 steps. As is common

practice, we output the probabilities generated in the last step instead of the sampled binary values, so that the result is a normal greyscale image. We display the resulting samples in Figures 1 and 2 (for reference, see randomly sampled training datapoints in Appendix F): we note that the model successfully generates samples with diversity, as in Figure 1 the model generates handbags both with and without handles, and in Figure 2 it renders both common styles for drawing the number 4.

It is clear that the model fails to generate as detailed of patterns exhibited in real FashionMNIST images since in our training algorithm, we represent a gray pixel as a random combination of black and white, so a checkerboard pattern of black and white and a patch of grey are not well-distinguished. We do this to ensure that our setup is comparable to classic RBM training [1]. It is potentially possible to fix this by adding spins over larger alphabets (e.g. real-valued) to the model.

## Broader Impact

We believe our work will be of most use to other researchers working on sparse graphical models with latent variables. We do not expect our research to disadvantage any individual. As with most machine learning tools, the proposed algorithm for classification could possibly fit to existing biases in the data. In fact, since our algorithm also learns per class distributions, a practitioner can sample from the distribution to further evaluate any biases implicitly modelled. Any practitioner using our method will need to apply the same due diligence as if they were fitting their data using a different method, such as logistic regression.

## Funding Information

This work was done in part while the authors were visiting the Simons Institute for the Theory of Computing for the Summer 2019 program on the Foundations of Deep Learning. A substantial part of the work was done while SG was a graduate student at UT Austin.

SG was supported by the JP Morgan AI PhD Fellowship. FK is supported in part by NSF award CCF-1453261 and Ankur Moitra's Packard Foundation Fellowship. AK is supported by NSF awards CCF-1909204 and CCF-1717896.

## Footnotes

[1]For the readers convenience, we include a self-contained derivation of (1) from (2) in Appendix B.1.

[2]Interestingly, recent work [25] has shown in a special case connections between the implicit bias of gradient descent in feedforward networks and $\ell_1$ regularization in function space.

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
