[Supplementary Material 1 · comp.pdf]

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

# A    Outline of the Appendix

Here we briefly outline the contents of each remaining section; each bold heading in the text below corresponds to a new section.

**Appendix B. Connections between Distribution Learning and Prediction in RBMs**    In this section we show that if you have learned the distribution of an RBM, then you have also in principle learned how to predict the output of corresponding feedforward networks. These feedforward networks are induced from a "self-supervised" prediction task: predicting the spin at node $i$ given observations of all other spins. This connection leverages a classical observation in probabilistic inference: inference in all tree-structured graphical models has an exact solution known as Belief Propagation (see e.g. [22, 21]); perhaps surprisingly, this observation is useful even though the RBM itself is not tree structured. Conversely, in the next subsection we give quantitative bounds showing that sufficiently good predictors for this self-supervised objective for every node $i$ allows us to recover the distribution of the corresponding RBM.

**Appendix C. Guarantees for Learning Feedforward Networks (with arbitrary distribution).** In this section we prove upper and lower bounds for learning one-layer feedforward networks with $f_\beta$ activations in the hidden units and inputs $X$ drawn from an arbitrary distribution such that $\|X\|_\infty \leq 1$.

In the first two subsections, we prove the needed approximation-theoretic results about our class of activations $f_\beta$, giving approximation results with uniform guarantees over the entire interval $\beta \in [0, 1]$. In the special case of $\beta = 0$, $f_\beta = \tanh$ and the needed result has essentially already been proved in the work of [23]. As explained in the first subsection, by a classical result of Bernstein (Theorem 7 below) it turns out that analyzing approximation theory for functions analytic on $[-1, 1]$ is equivalent to analyzing the function's extension into the complex plane. We develop the needed complex-analytic estimates (which crucially are uniform in $\beta$) in the following subsection. We note that the authors of [23] did not use Bernstein's result to prove their bound; their analysis of the $\beta = 0$ case is longer because they more or less reproduce the steps from the proof of the upper bound of Bernstein's Theorem.

After solving the approximation-theoretic question, we use them in an $\ell_1$-regression based algorithm for learning feedforward networks, using an explicit polynomial feature map and the logistic version of the Lasso with its corresponding nonparametric generalization bounds. We derive the needed $\ell_1$-norm bound in a clean way from the approximation-theoretic results using in part a Lemma of [46], previously used in [31]. This proves Theorem 2. In the last subsection, we prove that this result is nearly optimal under the hardness of sparse parity with noise, even in the case of $\tanh$ networks, using two different ways to construct a parity out of $\tanh$ units: one is a well-known construction from [47], the other is based on Taylor series expansion and is related to the MRF-to-RBM embedding result established in [18].

**Appendix D. Learning RBMs by Learning Feedforward Networks.**    In this section, we show how to derive structure recovery results (i.e. recovery of Markov blankets) for RBMs by using the feedforward network learning results developed in the previous section. Assuming $\eta$-nondegeneracy, we show how to learn the structure of the network by doing simple regression tests, e.g. comparing the minimal logistic loss achieved predicting node $i$ from all other nodes to the loss when node $j$ is excluded from the input. This proves Theorem 4. We explain in more detail in Remark 5 how this result is a significant improvement over previous results in interesting regimes where we know that the RBM can actually be sampled from in polynomial time. Based on this, we prove a result for learning the distribution: by Theorem 4 this reduces to the case where the structure is known, so by proving a good estimate (Lemma 12) on the convergence of the natural predictor of $X_i$ given its neighbors, the empirical conditional expectation and using the tools developed in Section B.3 gives the result. A key point here is that the empirical conditional expectation converges at a much faster rate than e.g. relying on Theorem 10, which gives better sample complexity guarantees.

Finally, we again prove some computational hardness results. We establish that the algorithm's dependence is essentially optimal in terms of $\eta$ and $\|w\|_1$ by using the Taylor-series based sparse parity construction from [18], related to the construction used above for $\tanh$ networks. For the dependence on $\lambda_2$, the hidden unit $\ell_1$-norm, we use a third, different construction of parity from [48]

for the RBM setting; this construction is not amenable to adding noise, but we are able to prove a lower bound on the runtime in terms of $\lambda_2$ for all SQ (Statistical Query) algorithms (see e.g. [49]).

**Appendix E. Learning a Feedforward Network by Learning RBMs.**   In this section, we prove Theorem 6, which lets us learn to predict in supervised RBMs under a natural conditional ferro-magneticity condition in a provably more computationally efficient way than applying distribution-agnostic methods for learning feedforward networks like Theorem 2. In Remark 8 we give a simple example where the gap is provable and explain the (in this case) simple intuition as to how the approach of Theorem 6 uses the structure of the input data in a favorable way.

The idea of this learning algorithm is essentially to use Bayes rule to reduce computing the posterior on the label (i.e. $\Pr(Y|X)$) to computing the conditional likelihood of the observed $X$ under the two possible values of the label. In some situations where the conditional law of $Y|X$ is very simple, this approach may be overkill as it requires to model the law of $X$; however, we are interested in the setting where the label $Y$ may have a large, complicated effect on $X$ so this approach seems perfectly reasonable. An obvious issue with using Bayes rule in this way is that even if the the RBM is already known perfectly, computing the normalizing constant for the conditional distribution under $Y = +$ or $Y = -$ in such a model is #BIS-Hard [50]. Fortunately, for our application we show that we can estimate the needed ratio of normalizing constants from the data using a simple variant of logistic regression.

What remains is to learn how to estimate the conditional log-likelihoods i.e. $\Pr(X|Y)$. Fortunately, even though under our assumptions the original RBM was not ferromagnetic, the conditional models we get by applying Bayes rule are indeed ferromagnetic so we can apply the methods developed in [19] for learning such a model. Here we need the results of [19] and not the earlier work of [18] as we expect the external fields in the resulting model to be inconsistent (have differing signs depending on the site). Once the structure is recovered, we can learn the coefficients of the log-likelihood using the results established in the previous section based on fast convergence of the empirical condition expectation, and using these coefficients we can accurately estimate $\Pr(X|Y)$ for the application of Bayes rule.

**Appendix F. Additional Experimental Data.**   In this section we include reference images from both datasets along with samples generated by our algorithm trained on MNIST.

## B  Connections between Distribution Learning and Prediction in RBMs

To our knowledge, Theorem 1 has not been previously noted in the literature on RBMs. However, this is not the first time connections between RBMs and message passing algorithms for inference has been investigated: for example, the work of [51] extensively studied the use of message passing algorithms (i.e. Belief Propagation and related algorithms) for estimating the mean and covariance matrix of nodes in an RBM, and the work of [44] used the related TAP approximation to derive better alternatives to constrastive divergence for training RBMs in practice. The key conceptual difference is that in these works, their goal is to solve a much harder problem (e.g. estimating marginals and $\log Z$) which is well-known to be NP-hard in general. In contrast, for our application to learning the relevant task ends up being predicting one node from the others, which it turns out is *not* computationally difficult if we know the model — conditioning on the other nodes breaks all cycles in the graph, which is the obstacle that makes inference difficult in general.

### B.1  Conditional Law Derivation

In this Appendix we give, for the reader's convenience, a self-contained derivation of the conditional law (1) described in Theorem 1 for $\mathbb{E}[X_i|X_{\sim i}]$ from (2). As described in the proof of the Theorem, the result is obtained as a special case of the Belief Propagation algorithm as described in a number of references, including [21, 22], which is derived by performing a more general version of this calculation. First recall that the joint conditional law on $X_i, H$ condiditioned on $X_{\sim i}$ is given by (2):

$$\Pr(X_i = x_i, H = h | X_{\sim i} = x_{\sim i}) \propto \exp\left(x_i(b_i^{(1)} + \sum_j W_{ij}h_j) + \langle W_{\sim i}^t x_{\sim i} + b^{(2)}, h\rangle\right).$$

The computation proceeds by rewriting this measure with respect to a "cavity" measure where all terms involving $X_i$ are removed. For each hidden unit $j$, define a corresponding probability measure

$$\mu_{H_j \to X_i}(h_j) \propto \exp\left(\sum_{k \neq i} W_{kj} x_j h_j + b_j^{(2)} h_j\right)$$

under which $\sum_j h_j \mu_{H_j \to X_i}(h_j) = \tanh(\sum_k W_{kj} x_j + b_j^{(2)})$ and rewrite the joint probability over $X, H$ as

$$\Pr(X_i = x, H = h | X_{\sim i} = x_{\sim i}) \propto \exp\left(x_i(b_i^{(1)} + \sum_j W_{ij} h_j)\right) \prod_j \mu_{H_j \to X_i}(h_j).$$

Now we compute that

$$\Pr[X_i = x_i | X_{\sim i} = x_{\sim i}]$$
$$= \sum_h x_i \Pr(X_i = x_i, H = h | X_{\sim i} = x_{\sim i})$$
$$\propto \sum_h \exp\left(x_i(b_i^{(1)} + \sum_j W_{ij} h_j)\right) \mu_{H \to X_i}(h)$$
$$= \exp(x_i b_i^{(1)}) \prod_{j=1}^{n_2} (\cosh(W_{ij}) + \sinh(x_i W_{ij}) \tanh(\sum_{k \neq i} W_{kj} x_j + b_j^{(2)}))$$
$$\propto \exp(x_i b_i^{(1)}) \prod_{j=1}^{n_2} (1 + x_i \tanh(W_{ij}) \tanh(\sum_{k \neq i} W_{kj} x_j + b_j^{(2)}))$$
$$= \exp\left(x_i b_i^{(1)} + \sum_{j=1}^{n_2} \log(1 + x_i \tanh(W_{ij}) \tanh(\sum_{k \neq i} W_{kj} x_j + b_j^{(2)}))\right)$$

where we used $\propto$ to ignore constants of proportionality independent of $x_i$ and in the third line we used Lemma 1 below. Therefore if we use that

$$\log(1 + \beta x_i) = \frac{1}{2} \log\frac{1 + \beta x_i}{1 - \beta x_i} + \frac{1}{2}(\log(1 + \beta x_i) + \log(1 - \beta x_i)) = \tanh^{-1}(\beta x_i) + \frac{1}{2}(\log(1 + \beta) + \log(1 - \beta))$$

where we see the last term does not depend on $x$, we can compute that

$$\mathbb{E}[X_i = x_i | X_{\sim i} = x_{\sim i}]$$
$$= \frac{\sum_{x_i} x_i \exp\left(x_i b_i^{(1)} + \sum_{j=1}^{n_2} \log(1 + x_i \tanh(W_{ij}) \tanh(\sum_{k \neq i} W_{kj} x_j + b_j^{(2)}))\right)}{\sum_{x_i} \exp\left(x_i b_i^{(1)} + \sum_{j=1}^{n_2} \log(1 + x_i \tanh(W_{ij}) \tanh(\sum_{k \neq i} W_{kj} x_j + b_j^{(2)}))\right)}$$
$$= \frac{\sum_{x_i} x_i \exp\left(x_i b_i^{(1)} + \sum_{j=1}^{n_2} x_i \tanh^{-1}(\tanh(W_{ij}) \tanh(\sum_{k \neq i} W_{kj} x_j + b_j^{(2)}))\right)}{\sum_{x_i} \exp\left(x_i b_i^{(1)} + \sum_{j=1}^{n_2} x_i \tanh^{-1}(\tanh(W_{ij}) \tanh(\sum_{k \neq i} W_{kj} x_j + b_j^{(2)}))\right)}$$
$$= \tanh\left(b_i^{(1)} + \sum_{j=1}^{n_2} \tanh^{-1}(\tanh(W_{ij}) \tanh(\sum_{k \neq i} W_{kj} x_j + b_j^{(2)}))\right)$$

where in the final step we used that $\tanh(z) = \frac{e^z - e^{-z}}{e^z + e^{-z}}$. From this we get (1) by plugging in the definition of $f_{\beta_{ij}}$.

**Lemma 1.** *For any $z \in \mathbb{R}$ we have the formula for moment generating function of a recentered Bernoulli:*

$$\mathbb{E}_{X \sim Ber_{\pm}(\tanh(z))}[\exp(\lambda X)] = \cosh(\lambda) + \sinh(\lambda) \tanh(z)$$

*where $Ber_{\pm}(\mu)$ denotes the distribution of a $\{\pm 1\}$-valued random variable with mean $\mu$.*

*Proof.* First recall that $\mathbb{E}_{X \sim Rad}[\exp(\lambda X)] = \cosh(\lambda)$ and $\mathbb{E}_{X \sim Rad}[X \exp(\lambda X)] = \tanh(\lambda)$. Therefore

$$
\begin{aligned}
\mathbb{E}_{X \sim Ber_{\pm}(\tanh(z))}[\exp(\lambda X)] &= \mathbb{E}_{X \sim Rad}\left[ e^{\lambda X} \frac{e^{zX}}{\cosh(z)} \right] \\
&= \frac{\cosh(z + \lambda)}{\cosh(z)} \\
&= \frac{\cosh(z)\cosh(\lambda) + \sinh(z)\sinh(\lambda)}{\cosh(z)} \\
&= \cosh(\lambda) + \sinh(\lambda)\tanh(z).
\end{aligned}
$$

$\square$

## B.2  2-layer Tanh Neural Network as Bayes-Optimal Prediction in an RBM

In particular, (1) lets us realize *any* standard 2-layer $\tanh$ neural network as the Bayes-optimal predictor in an RBM in a natural limit where the number of hidden neurons goes to infinity, but the effect of each hidden neuron is very small, so that the $\ell_1$ norm of the weights going into the top neuron stays bounded by a constant. Each hidden unit in the neural network corresponds in a direct way to several duplicated hidden units in the RBM. The construction is given explicitly in the next Lemma; we will not use the statement explicitly but use it to develop intuition for (1).

**Lemma 2.** *Suppose that* $g(x) = \tanh\left( u_0 + \sum_{j=1}^{T} u_j \tanh\left( M_{j0} + \sum_k M_{jk} x_k \right) \right)$ *where $x$ is $n$-dimensional, i.e. $g$ is a 2-layer neural network with* $\tanh$ *activations. Then*

$$
g(x) = \lim_{K \to \infty} \tanh\left( u_0 + \sum_{i=1}^{K} \sum_{j=1}^{T} \tanh(u_j/K) f_{|u_j/K|}\left( M_{j0} + \sum_k M_{jk} x_k \right) \right),
$$

*so by (1) from Theorem 1 the restriction of $f$ to $\{\pm 1\}^n$ is the Bayes-optimal predictor of a visible unit in an RBM with $n + 1$ total visible units where the activations of the other visible units are known.*

*Proof.* This follows from the observation in Remark 2 and from Theorem 1 by building the corresponding RBM with $KT$ hidden units. $\square$

## B.3  Distribution learning bounds from prediction bounds

In this section, we show how good estimates of the conditional prediction functions can be used in a direct way to recover the joint distribution of the RBM in total variation distance.

---
**Algorithm 2** DISTRIBUTIONFROMPREDICTORS
---
1: For every $i$ we suppose we are given $\hat{f}_i : \{\pm 1\}^n \to \mathbb{R}$ and set $\widehat{\mathcal{N}}(i)$ such that $\hat{f}_i$ is a predictor of node $i$ from other nodes that depends only on those in the set $\widehat{\mathcal{N}}(i)$
2: Define $\mathcal{S} := \{S : \exists i, S \subset \widehat{\mathcal{N}}(i)\}$
3: **for** $S \in \mathcal{S}$ **do**
4:     For all $i \in S$, define $\hat{w}_{S,i} := \mathbb{E}_{X \sim Uni(\{\pm 1\}^n)}[\tanh^{-1}(\hat{f}_i(X)) X_{S \setminus i}]$.
5:     Define $\hat{w}_S := \frac{1}{|S|} \sum_{i \in S} \hat{w}_{S,i}$.
6: Return the MRF with unnormalized pmf $\exp\left( \sum_{S \in \mathcal{S}} \hat{w}_S X_S \right)$.

---

**Lemma 3** ([26])**.** *Suppose $P, Q$ are distributions over random variable $X$ valued in $\{\pm 1\}^n$. If $P(x) \propto \exp(\sum_S p_S X_S)$ and $Q(x) \propto \exp(\sum_S q_S X_S)$ then*

$$
\mathbf{SKL}(P, Q) = \sum_S (p_S - q_S)(\mathbb{E}_P[X_S] - \mathbb{E}_Q[X_S]).
$$

*where* $\mathbf{SKL}(P, Q) = \mathbf{KL}(P, Q) + \mathbf{KL}(Q, P)$ *is the symmetrized KL divergence.*

*Proof.* From the definition we see

$$\mathbf{SKL}(P,Q) = \mathbb{E}_P\left[\log\frac{P(x)}{Q(x)}\right] - \mathbb{E}_Q\left[\log\frac{P(x)}{Q(x)}\right] = \mathbb{E}_P\left[\sum_S (p_S - q_S)X_S\right] - \mathbb{E}_Q\left[\sum_S (p_S - q_S)X_S\right]$$

so using linearity of expectation proves the result. $\qquad\square$

The following definition captures the level of contiguity $P$ has with the uniform measure when looking at small sets of coordinates.

**Definition 3.** *For any distribution $P$ on $\{\pm 1\}^n$ and $d \leq n$ we define*

$$\delta_P(d) := \inf_{|S|\leq d}\inf_{x_S} 2^{|S|}P(X_S = x_S).$$

**Lemma 4.** *For any function $f$ which depends on at most $d$ coordinates,*
$$\mathbb{E}_P[f(X)^2] \geq \delta_P(d)\mathbb{E}_{X\sim\{\pm 1\}^n}[f(X)^2]$$

The following Lemma is a standard observation used in most previous works on learning Ising models including [6, 7, 8] and others.

**Lemma 5.** *A $(\lambda_1, \lambda_2)$-bounded RBM satisfies $\delta_P(d) \geq (1 - \tanh(\lambda_1))^d$.*

*Proof.* In the $d = 1$ case this follows from the law of total expectation as $\mathbb{E}[X_i|H, X_{\sim i}] = \tanh(b_i^{(1)} + \sum_j W_{ij}H_j)$ and the term inside the $\tanh$ has magnitude at most $\lambda_1$ by definition. For general $d$ the result follows by induction, by using the above argument for a single spin and then applying the induction hypothesis to the model where than spin is plus and where that spin is minus, since these models are also $(\lambda_1, \lambda_2)$-bounded RBMs. $\qquad\square$

**Lemma 6.** *Let $\hat{P}$ denote the distribution returned by Algorithm* DISTRIBUTIONFROMPREDICTORS *and let $P$ be the true distribution. Let $\log P(x) = \sum_S w_S x_S$ and $\log\hat{P}(x) = \sum_S \hat{w}_S x_S$ be the Fourier expansions of the log-likelihoods. Then*

$$\mathbf{SKL}(\hat{P}, P) \leq \sum_S |w_S - \hat{w}_S|$$

$$\leq \sum_i \frac{2^{|\mathcal{N}(i)|/2+1}}{\sqrt{\delta_P(|\mathcal{N}(i)\cup\widehat{\mathcal{N}}(i)|)}}\sqrt{\mathbb{E}_{X'}[(\tanh^{-1}(\hat{f}_i(X')) - \tanh^{-1}(\mathbb{E}_P[X_i|X_{\sim i}]))^2]}$$

*where $X' \sim Uni(\{\pm 1\}^n)$.*

*Proof.* Define $w_S$ to be the true coefficient in the true MRF potential. By Lemma 3 and Holder's inequality we know $\mathbf{SKL}(P, \hat{P}) \leq 2\sum_S |\hat{w}_S - w_S|$. Then by Jensen's inequality and the Cauchy-Schwarz inequality,

$$\sum_S |\hat{w}_S - w_S| \leq \sum_S \frac{1}{|S|}\sum_{i\in S}|\hat{w}_{S,i} - w_S|$$

$$= \sum_i \sum_{S:i\in S}\frac{1}{|S|}|\hat{w}_{S,i} - w_S|$$

$$\leq \sum_i 2^{|\mathcal{N}(i)|/2}\sqrt{\sum_{S:i\in S}(\hat{w}_{S,i} - w_S)^2}.$$

Now using Plancherel's theorem [52], the fact that $f_i(x) = \tanh\left(\sum_{S:i\in S} w_S x_{S\setminus\{i\}}\right)$, and the definition of $\delta_P(d)$ gives the result. $\qquad\square$

## C Guarantees for Learning Feedforward Networks (with Arbitrary Distribution)

In this section we prove upper and lower bounds for learning one-layer feedforward networks with $f_\beta$ activations in the hidden units and inputs $X$ drawn from an arbitrary distribution such that $\|X\|_\infty \leq 1$.

## C.1 Preliminaries: Optimal Approximation of Analytic Functions

Identify $\mathbb{C}$ with $\mathbb{R}^2$ by taking $x$ to be real and $y$ to be the imaginary component of a complex number $z$. Define $\mathcal{E}_\rho$ to be the region bounded by the ellipse in $\mathbb{C} = \mathbb{R}^2$ centered at the origin with equation $\frac{x^2}{a^2} + \frac{y^2}{b^2} = 1$ with semi-axes $a = \frac{1}{2}(\rho + \rho^{-1})$ and $b = \frac{1}{2}|\rho - \rho^{-1}|$; the focii of the ellipse are $\pm 1$. In the present context, this is sometimes referred to as a *Bernstein ellipse*. For an arbitrary function $f : [-1, 1] \to \mathbb{R}$, let $E_D(f)$ denote the error of the best polynomial approximation of degree $D$ in infinity norm on the interval $[-1, 1]$ of $f$, i.e.

$$E_D(f) := \min_{P : \deg(P) \leq D} \max_{x \in [-1,1]} |f(x) - P(x)|. \tag{3}$$

The following theorem of Bernstein exactly characterizes the asymptotic rate at which $E_D(f)$ shrinks:

**Theorem 7** (Theorem 7.8.1, [53]). *Let $f$ be a function defined on $[-1, 1]$. Let $\rho_0$ be the supremum of all $\rho$ such that $f$ has an analytic extension on the interior of $\mathcal{E}_\rho$. Then*

$$\limsup_{D \to \infty} \sqrt[D]{E_D(f)} = \frac{1}{\rho_0}$$

*where we interpret the rhs as $\infty$ when $\rho_0 = 0$.*

For the definition of what it means for the function to be analytic on a region of the complex plane, we refer to a text on complex analysis such as [54]. For our application we need only the upper bound and we need a quantitative estimate for finite degree $d$. In the proof of the upper bound in [53], the following result is proved:

**Theorem 8** (Quantitative Variant of Theorem 7.8.1, [53]). *Suppose $f$ is analytic on the interior of $\mathcal{E}_{\rho_1}$ and $|f(z)| \leq M$ on the closure of $\mathcal{E}_{\rho_1}$. Then*

$$E_D(f) \leq \frac{2M}{\rho_1 - 1} \rho_1^{-D}.$$

This quantitative variant was previously used in [55] as part of a construction of low-degree approximations to the ReLU activation with specific properties. Note that when applying this theorem, we should center $f$ so that the constant $M$ is small, since adding constants to $f$ will obviously not change $E_d(f)$.

## C.2 Approximation Guarantees for $f_\beta$ Family of Activations

Recall that the activations $f_\beta$ were defined in Theorem 1 to be $f_\beta(x) = \frac{1}{\beta} \tanh^{-1}(\beta \tanh(x))$. Recall that if $\beta = 1$ then $f_\beta(x) = x$ so the function is analytic everywhere on $\mathbb{C}$, and if $\beta = 0$ is is $\tanh$ so it is meromorphic. For the remaining values of $\beta \in (0, 1)$, the function $f_\beta$ is slightly more complicated (it has branch cuts), however we show it is still nicely behaved near the real line.

**Lemma 7.** *For $\beta \in [0, 1]$ the function $f_\beta$ is analytic on the strip $\{x + iy : |y| < \pi/2\}$.*

*Proof.* Observe that

$$f_\beta'(z) = \frac{1 - \tanh^2(z)}{1 - \beta^2 \tanh^2(z)}..$$

Since $\tanh$ is analytic except at points of the form $z = \frac{\pi}{2}i + \pi k i$, the only other possible poles are solutions to $\beta^2 \tanh^2(z) = 1$, i.e. solutions to $\tanh(z) = \pm 1/\beta$. Recalling that $\tanh^{-1}(z) = \frac{1}{2}(\log(1 + z) - \log(1 - z))$ and taking into account the branch cut from $(-\infty, 0]$ for the logarithm, we see that the solutions to $\tanh(z) = 1/\beta$ are of the form

$$z = \frac{1}{2} \log \frac{1 + 1/\beta}{1/\beta - 1} + \frac{\pi i}{2} + k\pi i$$

and for $\tanh(z) = -1/\beta$ of the form

$$z = \frac{1}{2} \log \frac{1/\beta - 1}{1 + 1/\beta} + \frac{\pi i}{2} + k\pi i$$

for $k \in \mathbb{Z}$. In particular we see that $f_\beta'$ is analytic on the strip $\{x + iy : |y| < \pi/2\}$ so $f_\beta$ is as well (since the region is simply connected, this can be proved by path integration [54]). $\qquad\square$

To get a quantitative upper bound we will need to bound (the centered version of) $f_\beta$ on the Bernstein ellipse, which will require us to back away from the singularities of $f'_\beta$ on the lines $y = \pm\pi/2$. The following Lemma proves that $f'_\beta$ is uniformly bounded in a slightly smaller region:

**Lemma 8.** *For all $\beta \in [0,1]$, $|f'_\beta(z)| \leq 2$ everywhere on the closed strip $\{x + iy : |y| \leq \pi/4\}$.*

*Proof.* Observe that

$$f'_\beta(z) = \frac{1 - \tanh^2(z)}{1 - \beta^2 \tanh^2(z)} = \frac{\cosh^2(z) - \sinh^2(z)}{\cosh^2(z) - \beta^2 \sinh^2(z)}$$

$$= \frac{1}{1 + (1 - \beta^2)\sinh^2(z)} = \frac{1}{1 + (1 - \beta^2)\frac{\cosh(2z)-1}{2}}$$

using the identies $\cosh^2(x) - \sinh^2(x) = 1$ and $\sinh^2(z) = \frac{\cosh(2z)-1}{2}$. Since $\cosh(2x + 2iy) = \frac{e^{2x+2iy}+e^{-2x-2iy}}{2}$ we see that under the assumption $|y| \leq \pi/4$ that $\cosh(2x + 2iy)$ lies in the right half plane, therefore $|1 + (1 - \beta^2)\frac{\cosh(2z)-1}{2}| \geq |1 - (1 - \beta^2)/2| \geq 1/2$ which proves the result. $\square$

**Lemma 9.** *For any $\beta \in [0,1]$, arbitrary $h \in \mathbb{R}$, and any $R \geq 0$,*

$$E_D(f_\beta(Rx + h)) \leq \frac{4R(1 + 2R)}{(1 + 1/2R)^D}$$

*Proof.* Just for this proof define $g_{\beta,h}(x) := f_\beta(Rx+h) - f_\beta(h)$. We prove this bound by application of Bernstein's theorem. By Lemma 7 we know that $f_\beta$ is analytic on the strip $\{x + iy : |y| < \pi/2\}$ so in particular it is analytic on the closed strip $\{x + iy : |y| \leq \pi/4\}$, and by Lemma 8 we know that $|f'_\beta| \leq 2$ on the closed strip.

We now compute $\rho$ so that $R\mathcal{E}_\rho$ is contained in the latter strip. We solve

$$\frac{1}{2}(\rho - \rho^{-1}) = \frac{\pi}{4R}$$

which gives $\rho^2 - \frac{\pi}{2R}\rho - 1 = 0$ so $\rho = \frac{\pi/2R + \sqrt{\pi^2/4R^2 + 4}}{2} > 1 + 1/2R$. Since $|g'_{\beta,h}(z)| \leq R|f'_\beta| \leq 2R$ on the closure of the ellipse, it follows by the mean-value theorem that $|g_{\beta,h}| \leq 2(1 + 1/2R)R \leq 1 + 2R$ on $\mathcal{E}_{1+1/2R}$ and applying Theorem 8 gives the result. $\square$

### C.3 Learning Feedforward Networks under $\ell_\infty$ Bounded Input

Since the final activation in our network is $\tanh$, we recall some useful facts about logistic regression and the logistic loss which we will use.

**Definition 4.** *The* logistic loss *is defined to be*

$$\ell(v, y) := \log(1 + e^{-2vy}).$$

*We note that the factor of 2 in the exponent and the normalization differ depending on convention.*

The following facts about the logistic loss which can be checked from the definition (or see a reference such as [56]):

**Fact 1.** *The following are true if $y \in \{\pm 1\}$ is fixed:*

1. *$\ell(v, y)$ is convex and 2-Lipschitz in $v$.*

2. *$\ell(v, y) = -\log \Pr(\hat{Y} = y)$ where $\hat{Y}$ is a $\{\pm 1\}$-valued random variable with expectation $\tanh(v)$.*

3. *$\frac{\partial}{\partial v}\ell(v, y) = \frac{-2ye^{-2vy}}{1 + e^{-2vy}}$ and $\frac{\partial^2}{\partial v^2}\ell(v, y) = \frac{2}{1 + \cosh(2v)}$.*

*Furthermore if $Y$ is a $\{\pm 1\}$-valued random variable (and $v$ is deterministic) then*

4. $\mathbb{E}_Y \ell(v, Y) = \mathbf{KL}(\mathcal{L}(Y), \mathcal{L}(\hat{Y})) + H(Y)$ where $\hat{Y}$ is defined above, $\mathcal{L}(Y)$ denotes the law of random variable $Y$, $\mathbf{KL}$ denotes the Kullback-Liebler divergence and $H$ denotes the Shannon entropy.

We recall the following Theorem which states the agnostic learning guarantee for fitting $\ell_1$-constrained predictors in logistic loss, i.e. the logistic version of the Lasso:

**Theorem 9** (Theorem 26.15 of [56]). *Suppose that $X$ is a random vector in $\mathbb{R}^n$ such that $\|X\|_\infty \leq 1$ almost surely and $Y$ is an arbitrary $\{\pm 1\}$-valued random variable. Then with probability at least $1 - \delta$, simultaneously for all $w$ with $\|w\|_1 \leq R$ it holds that*

$$\hat{\mathbb{E}}[\ell(w \cdot X, Y)] \leq \mathbb{E}[\ell(w \cdot X, Y)] + 4R\sqrt{\frac{2\log(2n)}{m}} + 2R\sqrt{\frac{2\log(2/\delta)}{m}}$$

*where $\hat{\mathbb{E}}$ denotes the empirical expectation over $m$ i.i.d. copies $(X_1, Y_1), \ldots, (X_m, Y_m)$ of $(X, Y)$.*

In order to bound the $\ell_1$ norm of our predictor we will need the following Lemmas:

**Lemma 10** ([46], Lemma 2.13 of [31]). *Suppose $p(x) = \sum_{i=0}^D \beta_i x$ and $|p(x)| \leq M$ for $x \in [-1, 1]$, then $\sum_{i=0}^D \beta_i^2 \leq (D+1)(4e)^{2D} M^2$.*

**Lemma 11.** *Suppose that $p(x) = \sum_{i=0}^D a_i (w \cdot x)^i = \sum_\alpha u_\alpha x^\alpha$. Then*

$$\sum_\alpha |u_\alpha| \leq \sqrt{\sum_i a_i^2 (1 + \|w\|_1)^D}.$$

*Proof.* For any multi-index $\alpha$ let $w_\alpha := \prod_{i \in \alpha} w_i$ and observe by the multinomial theorem

$$p(w \cdot x) = \sum_i a_i (w \cdot x)^i = \sum_i a_i \sum_{|\alpha|=i} \binom{i}{\alpha} w_\alpha x^\alpha.$$

Therefore by the triangle inequality, multinomial theorem, and Cauchy-Schwarz inequality

$$\sum_\alpha |u_\alpha| \leq \sum_i |a_i| \sum_{|\alpha|=i} \binom{i}{\alpha} |w_\alpha| = \sum_i |a_i| \|w\|_1^i \leq \sqrt{\sum_i a_i^2 \sum_i \|w\|_1^{2i}} \leq \sqrt{\sum_i a_i^2 (1 + \|w\|_1)^d}$$

where in the last step we used $1 + x^2 + x^4 + \cdots + x^k \leq (1+x)^k$ for $x \geq 0$. $\square$

**Theorem 10.** *Suppose that $Y$ is a random variable valued in $\{\pm 1\}$, $X$ is a random vector such that $\|X\|_\infty \leq 1$ almost surely and*

$$\mathbb{E}[Y|X] = \tanh\left(b^{(1)} + \sum_j w_j f_{\beta_j}\left(b_j^{(2)} + \sum_k W_{jk} X_k\right)\right)$$

*where $b^{(1)} \in \mathbb{R}$, $\beta_j \in [0, 1]$, $w$ is an arbitrary real vector and $W$ is an arbitrary real matrix. Let $W_j$ denote column $j$ of $W$. Then $\ell_1$-constrained regression on the degree $D$ monomial feature map $\varphi_D(x) \mapsto \left(\prod_{i \in S} X_i\right)_{|S| \leq d}$ with $\ell_1$ constraint*

$$\|w\|_1 \leq R := |b^{(1)}| + \sqrt{D+1}(4e)^{D+1} \sum_j |w_j|(1 + \|W_j\|_1)^{D+1}$$

*returns a predictor $\hat{w}$ such that with probability at least $1 - \delta$,*

$$\mathbb{E}[\ell(\hat{w} \cdot \varphi_d(X), Y)] - \mathbb{E}[\ell(v^*(X), Y)]$$

$$\leq 8 \sum_j |w_j| \frac{\|W_j\|_1 + 2\|W_j\|_1^2}{(1 + 2/\|W_j\|_1)^D} + 4R\sqrt{\frac{2D\log(2n)}{m}} + 2R\sqrt{\frac{2\log(2/\delta)}{m}}$$

*where $v^*(X) := \tanh^{-1}(\mathbb{E}[Y|X]) = b^{(1)} + \sum_j w_j f_{\beta_j}\left(b_j^{(2)} + \sum_k W_{jk} X_k\right)$ is the minimizer of the expected logistic loss over all measurable functions of $X$. The runtime is $poly(n^D)$.*

*Proof.* The fact that $v^*(X)$ is the minimizer of the logistic loss $\mathbb{E}[\ell(h(X),Y)]$ over all $X$-measurable functions $h$ can be seen from Fact 1. To derive the bound we combine the approximation-theoretic guarantees developed in the previous section with the $\ell_1$ guarantee for logistic Lasso.

For the approximation step, define $w^*$ so that $w^* \cdot \varphi_d(X)$ is given by replacing each activation $f_{\beta_j}$ by its best polynomial approximation $P_j$ on the interval $[b_j^{(2)} - \|W_j\|_1, b_j^{(2)} + \|W_j\|_1]$. By the triangle inequality and Lemma 9, for any $x \in \{\pm 1\}^n$,

$$|v^*(x) - w^* \cdot \varphi_d(x)| \leq \sum_j |w_j| |(f_{\beta_j} - P_j)(b_j^{(2)} + \sum_k W_{jk} x_k)| \leq 4 \sum_j \frac{|w_j| \left(\|W_j\|_1 + 2\|W_j\|_1^2\right)}{(1 + 2/\|W_j\|_1)^D}.$$

Since the logistic loss is 2-Lipschitz (Fact 1.1), this implies that

$$\mathbb{E}[\ell(w^* \cdot \varphi_D(X), Y)] \leq \mathbb{E}[\ell(v^*(X), Y)] + 8 \sum_j \frac{|w_j| \left(\|W_j\|_1 + 2\|W_j\|_1^2\right)}{(1 + 2/\|W_j\|_1)^D}. \tag{4}$$

Combining Lemma 8, Lemma 10 and Lemma 11 and using the triangle inequality shows that $\|w^*\|_1 \leq R$ where $R$ is as specified in the Theorem statement. Then applying Theorem 9 and combining it with (4) gives the desired inequality bounding the error of the predictor $\hat{w}$. $\qquad\square$

To simplify usage of this Theorem, we give the following slightly less precise bound which will be used from now on:

**Corollary 1.** *In the same setting as Theorem 10, if we assume that $\|W_j\|_1 \leq \lambda$ for every $j$ and $\lambda \geq 2$, then with probability at least $1 - \delta$, $\mathbb{E}[\ell(\hat{w} \cdot \varphi_d(X), Y)] - \mathbb{E}[\ell(v^*(X), Y)] \leq \epsilon$ as long as the number of samples $m$ satisfies $m = \Omega((|b^{(1)}|^2 \lambda^{O(D)}) \log(2n/\delta))$ where $D = O(\lambda \log(\|w\|_1 \lambda/\epsilon))$ and the runtime of the algorithm is $poly(n^D)$.*

*Proof.* In order to make the first term of the bound on $\mathbb{E}[\ell(\hat{w} \cdot \varphi_d(X), Y)] - \mathbb{E}[\ell(v^*(X), Y)]$ at most $\epsilon/2$, we can upper bound it by $O(\|w\|_1 \lambda^2/(1 + 2/\lambda)^D)$ and see that it suffices to take $D = \Omega(\lambda \log(\|w\|_1 \lambda/\epsilon))$. Then $R = |b^{(1)}| + \exp(O(D))\|w\|_1 \lambda^{D+1} = |b^{(1)}| + \lambda^{O(D)}$ so it suffices to take $m = \Omega((|b^{(1)}|^2 + \lambda^{O(D)}) \log(2n/\delta))$ $\qquad\square$

**Remark 3.** *In the analysis of Theorem 10 we did not concern ourselves with the exact constants in the runtime. However, if we are interested in optimizing the runtime it should be noted that instead of getting a precise estimate of the empirical risk minimizer when computing the logistic regression, one can achieve a similar statistical guarantee by using a single pass of stochastic mirror descent/exponentiated gradient (see reference text [57]), e.g. as used in [8] where the needed high-probability guarantees can be found.*

### C.4   Nearly Matching computational lower bounds

In this section, we show that the runtime guarantee of Corollary 1 is close to optima: more precisely its runtime is optimal in $\|w\|_1$ and $\epsilon$ up to a $\log\log$ factor in the exponent, and also that at least sub-exponential dependence on $\lambda$ is required. We first recall the definition of this problem and a standard hardness assumption for learning sparse parity with noise. We phrase it in terms of a testing problem versus the uniform distribution, which is equivalent to a learning formulation (i.e. recovering $S$ below), by boosting the probability of success and using a standard reduction of removing one coordinate at a time and testing (see e.g. [20]).

**Definition 5.** *The $k$-sparse parity with noise* distribution *is the following distribution on $(X, Y)$ parameterized by $\eta \in (0, 1/2)$ and an unknown subset $S$ of size $k$:*

1. *Sample $X \sim \text{Unif}(\{-1, +1\}^n)$.*

2. *With probability $1/2 + \eta$, set $Y = \prod_{s \in S} X_s$, and with probability $1/2 - \eta$, set $Y = (-1)\prod_{s \in S} X_s$.*

*The $k$-sparse parity with noise* problem *is to test between the uniform and $k$-sparse parity with noise with sum of probability of Type I and Type II errors upper bounded by $0.01$, given access to an oracle which generates samples from one of the two distributions.*

**Assumption 1** (Hardness of learning sparse parity with noise). *Suppose $k_n$ is an arbitrary sequence of positive integers with $k_n = o(n^{1-\epsilon})$ for any $\epsilon > 0$ and $n$ growing, any algorithm which solve the $k$-sparse parity with noise testing problem must have runtime $n^{\Omega(k_n)}$.*

The reason for the condition $k_n = o(n^{1-\epsilon})$ is simply because the number of sets of size $n$ is $2^n$, not $n^n$, so small correction factors in the exponent are needed when $k$ is comparable to $n$. The best known algorithm for learning sparse parity with noise runs in time $n^{0.8k_n}$ [20].

**Theorem 11.** *In the setting of Corollary 1 and under Assumption 1, for $\lambda \leq 2$ there exists families of models (one with $\epsilon$ a constant, one with $\|w\|_1$ a constant) where a runtime of*

$$n^{\Omega\left(\frac{\log(\|w\|_1/\epsilon)}{\log\log(\|w\|_1/\epsilon)}\right)}$$

*is needed for any algorithm to achieve $\epsilon$ error with high probability, regardless of its sample complexity and even in the case of $\tanh$ activations ($\beta_j = 0$ for all $j$). There also exists a sequence of models with $\lambda = \Theta(n \log(n))$ and $\|w\|_1 = O(\sqrt{n})$ which requires runtime*

$$n^{\Omega(\sqrt{\lambda/\log^2(\lambda)\log(n)}\log\|w\|_1)}$$

*to achieve error $\epsilon = 0.01$ with high probability.*

*Proof.* We first show a lower bound of $n^{\Omega(\log(\|w\|_1/\epsilon))}$ for a family of models where $\lambda \leq 1$. Recall we are proving a lower bound in the $\beta_j = 0$ case where all activations are $\tanh$. The lower bound is shown by building a parity function out of $\tanh$ functions exactly using a simple taylor series expansion argument, under the assumption that the input to the network is in the hypercube $\{\pm 1\}^n$. The construction proceeds in a similar fashion to the sparse parity with noise lower bound for learning RBMs of bounded hidden degree established in [18]. We first describe the construction of a parity function on boolean inputs $x_1, \ldots, x_k$. It suffices to build this parity with a small (constant-size) coefficient, since we can repeat it to make the coefficient larger. We start from the fact that

$$\tanh(z) = 2 \sum_k \frac{(-1)^k}{\pi^{2k+2}} (1 - 1/4^{k+1}) \zeta(2k+2) z^{2k+1}$$

for $|z| < \pi/2$ and recall that the Riemann $\zeta$ function does not vanish on even integers [54], so every coefficient in this expansion is nonzero. Furthermore it is known that $\zeta(n) \to 1$ as $n \to \infty$, since this follows from the power series definition of $\zeta(s) = \sum \frac{1}{n^s}$, so we can write

$$\tanh(z) = \sum_k a_{2k+1} z^{2k+1}$$

where $a_{2k+1} \neq 0$ for any $k$ and $|a_{2k+1}| = \Theta(1/\pi^{2k+2})$. From this we can see that for some constant $c \neq 0$,

$$x_1 \cdots x_{2k+1} = c \frac{(2k+1)^{2k+1}}{a_{2k+1}} \tanh\left(\frac{x_1 + \cdots + x_{2k+1}}{2k+1}\right) + p(x)$$

where $p(x)$ is of degree at most $k - 1$, using that $x_i^2 = 1$ for all $i$ on the hypercube; here the constant $c$ (which is close to 1) is a fixed correction factor to handle the small effect of maximum-degree terms coming from expanding higher order terms in the $\tanh$ power series. We can inductively rewrite each of the highest-order coefficients of $p$ in terms of $\tanh$ and lower order monomials: this ultimately gives us a way to write parity as a linear combination of $\tanh$ functions. Using this, we can rewrite $\tanh(\frac{1}{4} x_1 \cdots x_{2k+1})$ as a two-layer $\tanh$ network with $\|w\|_1 = k^{O(k)}$ and $\lambda \leq 1$. Taking $\epsilon = 1/16$ and using the hardness of $k$-sparse parity with noise, we get that the runtime for learning the corresponding network is at least $n^{\Omega(k)} = n^{\Omega(\log(\|w\|_1)/\log\log(\|w\|_1))}$.

We can similarly prove a lower bound of $n^{\Omega(\log(1/\epsilon)/\log\log(1/\epsilon))}$ for constant $\lambda, \|w\|_1$ by using the same method to convert $\tanh(\eta x_1 \cdots x_{2k+1})$ into a two-layer network and by taking $\eta = k^{-\Theta(k)}$ so that the $\ell_1$ norm of the coefficients is shrunk to be at most 1. Taking $\epsilon = \Theta(\eta)$ and using the sparse parity with noise lower bound as above gives the result.

Finally, we give a lower bound showing exponential dependence on $\lambda$ is necessary. We use the well-known fact that a parity can be written as a small sum of threshold functions [47]. For $k$ even,

$$x_1 \cdots x_k = \mathbb{1}[x_1 + \cdots + x_k \geq -k] - 2(\mathbb{1}[x_1 + \cdots + x_k \geq -k+2] - \mathbb{1}[x_1 + \cdots + x_k \geq -k+4] + \cdots)$$

with a total of $k + 1$ terms in the sum on the rhs. We now consider replacing each threshold function with the approximation $\mathbb{1}[a \geq b] \approx \frac{1+\tanh(\lambda'(a-b+1/2))}{2}$ for some $\lambda' > 0$. Note that the error of this approximation for a singe threshold unit and integers $a, b$ is maximized when $a - b = 0$ where the error is $\frac{1-\tanh(\lambda'/2)}{2} = O(e^{-\lambda'})$. Therefore by Holder's inequality, the error in approximating $x_1 \cdots x_k$ by replacing all of the threshold functions is $O(ke^{-\lambda'}) = O(ke^{-\lambda/(k+1/2)})$, where we used that $\lambda = (k + 1/2)\lambda'$ where $\lambda$ is the hidden node $\ell_1$ norm as used previously. By adding a $\tanh$ nonlinearity on top of the approximate parity, this gives an approximate construction of sparse parity with noise.

Taking $k = \sqrt{n}$ and $\lambda = \Theta(k^2 \log(n))$ we see that the resulting model is **TV**-distance $n^{-\Theta(k)}$ from sparse parity with noise, so any algorithm with runtime $cn^{-\Theta(k)}$ cannot distinguish this model from sparse parity with noise with probability better than 75% for sufficiently small constant $c > 0$. From the assumed hardness of learning sparse parity with noise, any algorithm succeeding to distinguish this model from the uniform distribution with sufficiently small error probability requires runtime $n^{\Omega(k)} = n^{\sqrt{\lambda/\log^2(\lambda)\log(n)}\log\|w\|_1}$. $\qquad\qquad\square$

**Remark 4.** *In the second construction in the proof of Theorem 11, based off of approximating threshold functions, the computational lower bound becomes stronger if we allow the algorithm access to less data (recall that for a fixed noise level, $\Theta(k \log n)$ samples suffice information-theoretically for sparse parity with noise). If we only allow to use $\Theta(k \log n)$ samples as information-theoretically required, then we can take $\lambda = \Theta(k(\log k + \log\log n))$ and the runtime required is $n^k = n^{\lambda/(\log\log n + \log(\lambda))}$.*

# D  Learning RBMs by Learning Feedforward Networks

## D.1  Structure and Distribution Learning Guarantees

In this section we discuss application of the prediction guarantees from the previous section to structure and distribution learning. As motivation, recall that in undirected graphical models the *Markov blanket* or *neighborhood* of a node $i$, the minimal set of nodes which separate node $i$ from the rest of the model in the underlying graph, is one of the most interesting pieces of information to learn about a node. By the Markov property, node $i$ interacts directly only with nodes in its Markov blanket, in the sense that $X_i$ is conditionally independent of all other nodes $X_k$ given the values of nodes $X_j$ for all $j$ in the markov blanket of $i$. Learning the markov blanket of all nodes, equivalently learning the underlying graph of the Markov Random Field, is referred to as *structure learning*. It is also known (see e.g. [18]) that once we have performed structure learning, distribution learning (e.g. in total variation distance) becomes a conceptually straightforward task as it can typically be reduced to solving low-dimensional regression problems.

As explained in the introduction, learning the structure requires a non-degeneracy condition on neighbors (recall the definition of $\eta$-nondegeneracy from above). In the introduction, we stated that if all edges are $\eta$-nondegenerate then we can learn the structure perfectly; in the next Theorem, we state a slightly more precise result giving the result we can successfully test between non-neighbors and $\eta$-nondegenerate neighbors, without requiring nondegeneracy on the entire model. Since our guarantee holds with high probability, using the union bound it immediately gives a result for structure recovery under $\eta$-nondegeneracy.

**Theorem 12.** *Let $i$ and $j$ be two visible nodes in a $(\lambda_1, \lambda_2)$-bounded RBM. Let $H_0$ be the hypothesis that nodes $i$ and $j$ are not two-hop neighbors and $H_1$ the hypothesis that nodes $i$ and $j$ are $\eta$-nondegenerate two-hop neighbors. Given $\delta > 0$ and $m = \Omega(\lambda_2^{O(D)} \log(2n/\delta))$ i.i.d. samples where $D = O(\lambda_2 \log(\lambda_1\lambda_2/\eta))$, we can test in time $poly(n^D)$ between $H_0$ and $H_1$ with sum of Type I and Type II errors upper bounded by $\delta$.*

*Proof.* We run the following testing procedure:

1. Run the $\ell_1$ regression algorithm from Theorem 1 to predict $X_i$ from $X_{\sim i}$ and from $X_{\sim i,j}$.

2. Repeat the previous step with $i$ and $j$ reversed.

3. If the decrease in prediction accuracy for removing $i$ or $j$ is at least $3\eta/4$ in either step 1 or step 2, reject $H_0$.

That this works follows by combining Theorem 1 and Corollary 1, by choosing $\epsilon = \eta/8$ under $H_0$ the difference in prediction error is at most $2\epsilon$ whereas under $H_1$ it must be at least $\eta - 2\epsilon$. $\qquad\square$

Assuming that all 2-hop neighbors in the RBM are $\eta$-nondegenerate, the above Theorem lets us recover the structure of the RBM (its 2-hop neighborhoods) in time $poly(n^D)$. In the following remark, we explain how large $D$ is in the regimes where we know polynomial time sampling from the RBM is possible:

**Remark 5** (Comparison to polynomial time sampling regimes)**.** *Dobrushin's uniqueness criterion is probably the most well-known sufficient condition for sampling to be possible in polynomial time in a general pairwise model. Dobrushin's condition is that for every node $i$, the total $\ell_1$-norm of the edges touching node $i$ is at most 1, where the mixing time guarantees for Glauber dynamics become worse as the maximum norm approaches 1 (see [28]). This condition is tight in the example of the Ising model on the complete graph (Curie-Weiss), or for the bipartite complete graph (i.e. dense RBM) with all edge weights positive and equal and an equal number of visible and hidden units.*

*Under Dobrushin's uniqueness criterion on the RBM, we have that $\lambda_1, \lambda_2 \leq 1$ so $D = O(\log(1/\eta))$. As mentioned above, we cannot compute $\eta$ in terms of just the edge weights for general models, but if we for example assume the model is $d$-regular and has all edge weights equal to $+1/d$ and no external field then it is not too hard to show that $\eta = \Omega(1/d^2)$ (see e.g. [18]), so in this case the overall runtime is $n^{\log(d)}$. We expect that under Dobrushin's condition $\eta = \Omega(1/d^2)$ except in perhaps some rare degenerate situations. This means the runtime is improved by an exponential factor in the exponent compared to what one gets by just applying the RBM to MRF reduction, since learning $d$-wise MRFs is known to require $n^d$ time in general [8].*

*In some other interesting contexts, it is also known that polynomial time sampling can only be guaranteed when $\lambda_1, \lambda_2 = O(1)$: for antiferromagnetic Ising models on bounded degree graphs with equal edge weights the sharp result is known for every $d$ [58, 59, 29] and embedding these Ising models as RBMs with hidden nodes of degree 2 in a straightforward way gives models with $\lambda_1, \lambda_2 = O(1)$ and $\eta = \Omega(1/d^2)$ (see Example 1 above).*

For distribution learning we will need the following technical Lemma, which is proved in Appendix D.2 using the local Rademacher complexity framework [60]. Informally it says that if $X$ is a random variable with a density with respect to the uniform measure on $\{\pm1\}^n$ that is lower bounded by a constant, then given a number of samples $m$ which is large with respect to the size of the domain the natural estimator of $\tanh^{-1}(\mathbb{E}[Y|X])$ has error which converges at a $1/m$ rate, which generalizes the case of estimating the (exponential-family parameterization of) mean, the $n = 0$ case, in a natural way. Since the bound depends exponentially on $n$, we will only apply it in settings where we expect $n$ is small. Similar bounds are used in previous works including [5, 6] and proved using different methods, though they are not quite as optimized (e.g. deriving this result from Lemma 3.2 of [6] would give a $1/\gamma^2$ dependence); this bound can be shown to be optimal up to constants.

**Lemma 12.** *Suppose that $X$ is a random variable valued in $\{\pm1\}^n$ with $\Pr(X = x) \geq \gamma/2^n$ for every $x$ and $Y$ is a random variable valued in $\{\pm1\}$. Suppose that $|\mathbb{E}[Y|X]| \leq r$ for $r < 1$. Let $\hat{\mathbb{E}}[Y|X]$ be the empirical conditional expectation of $Y$ given $X$ based upon $m$ i.i.d. samples of $(X, Y)$ and define $h(X) := \min(\max(\mathbb{E}[Y|X], r), -r)$. Then with probability at least $1 - \delta$,*

$$\mathbb{E}[(\tanh^{-1}(h(X)) - \tanh^{-1}(\mathbb{E}[Y|X]))^2] \lesssim \frac{2^n/\gamma + \log(1/\delta)}{(1 - r^2)^2 m}$$

*where $\lesssim$ denotes inequality up to an absolute constant.*

We present the proof of this lemma in the subsequent subsection. From this Lemma we straightforwardly get the right result for learning a sparse RBM with known 2-hop neighborhoods.

**Lemma 13.** *For any $(\lambda_1, \lambda_2)$-bounded RBM where the maximum two-hop degree of any visible node is at most $d_2$ and where $\|b^{(1)}\|_\infty \leq B$, for $\delta > 0$ and $m = \Omega\left(n^2 \left(\frac{2}{(1 - \tanh(\lambda_1))}\right)^{d_2 + 1} \log(n/\delta)/\epsilon^4\right)$ we have that with probability at least $1 - \delta$, Algorithm* DISTRIBUTIONFROMSTRUCTURE *given $m$*

---

**Algorithm 3** DISTRIBUTIONFROMSTRUCTURE

1: We assume for every node $i$ we are given a recovered neighborhood $\widehat{\mathcal{N}}(i)$. $\widehat{\mathcal{N}}(i)$
2: For every node $i$ with neighborhood $\widehat{\mathcal{N}}(i)$, let $f_i(X) := \widehat{\mathbb{E}}[X_i | X_{\widehat{\mathcal{N}}(i)}]$ be the empirical conditional expectation of $X_i$ given $X_{\widehat{\mathcal{N}}(i)}$.
3: Return the output of Algorithm DISTRIBUTIONFROMPREDICTORS run with these $f_i$.

---

*samples and $\widehat{\mathcal{N}}(i) = \mathcal{N}(i)$ for every $i$ returns a distribution $\hat{P}$ which is $\epsilon$-TV close to the distribution of the RBM. Furthermore, if $w_S, \hat{w}_S$ are as defined as in Lemma 6 then*

$$2\mathbf{TV}(P, \hat{P})^2 \leq \mathbf{SKL}(P, \hat{P}) \leq \sum_S |w_S - \hat{w}_S| \leq \epsilon^2.$$

*Proof.* By Lemma 6, Lemma 5 and Lemma 12 we have

$$
\begin{aligned}
\mathbf{SKL}(\hat{P}, P) &\leq \sum_S |w_S - \hat{w}_S| \\
&\leq \sum_i \frac{2^{d_2/2+1}}{(1 - \tanh(\lambda_1))^{d_2/2}} \sqrt{\mathbb{E}_{X \sim Uni(\{\pm 1\}^n)}[(\tanh^{-1}(h_i(X)) - \tanh^{-1}(\mathbb{E}_P[X_i|X_{\sim i}])^2]} \\
&\leq \sum_i \frac{2^{d_2/2+1}}{(1 - \tanh(\lambda_1))^{d_2}} \sqrt{\mathbb{E}_{X_{\mathcal{N}(i)}}[(\tanh^{-1}(h_i(X)) - \tanh^{-1}(\mathbb{E}_P[X_i|X_{\sim i}])^2]} \\
&\leq \sum_i \frac{2^{d_2/2+1}}{(1 - \tanh(\lambda_1))^{d_2}} \sqrt{\frac{2^{d_2}/(1 - \tanh(\lambda_1))^{d_2} + \log(n/\delta)}{(1 - \tanh(\lambda_1)^2)^2 m}}
\end{aligned}
$$

and by Pinsker's inequality $\mathbf{TV}(\hat{P}, P)^2 \leq \mathbf{SKL}(\hat{P}, P)/2$ so the result follows. $\square$

**Theorem 13.** *Suppose that all visible nodes in an RBM which are neighbors in the Markov blanket sense are $\eta$-nondegenerate neighbors, and that maximum 2-hop degree of any visible node is at most $d_2$. Then given $\delta > 0$ and $m = \Omega(\lambda_2^{O(D)} \log(2n/\delta) + n^2 \left(\frac{2}{(1-\tanh(\lambda_1))}\right)^{d_2+1} \log(n/\delta)/\epsilon^4)$ i.i.d. samples where $D = O(\lambda_2 \log(\lambda_1 \lambda_2/\eta))$ samples, Algorithm DISTRIBUTIONFROMSTRUCTURE run with the set of $\eta$-nondegenerate neighbors output by Theorem 12 returns with probability at least $1 - \delta$ a distribution which is $\epsilon$-TV close to the true distribution of the RBM.*

*Proof.* This follows by combining Theorem 12 and Lemma 13. $\square$

**Remark 6.** *If we do not assume that all neighbors are $\eta$-nondegenerate, then by Theorem 16 it is impossible to get a nontrivial distribution learning guarantee assuming the hardness of learning sparse parity with noise, in the sense that the naive approach of forgetting the RBM structure entirely and using MRF learning results (e.g. [8]) cannot be improved.*

### D.2  Proof of Lemma 12

We recall the statement of Lemma 12. Suppose that $X$ is a random variable valued in $\{\pm 1\}^n$ with $\Pr(X = x) \geq \gamma/2^n$ for every $x$ and $Y$ is a random variable valued in $\{\pm 1\}$. Suppose that $|\mathbb{E}[Y|X]| \leq r$ for $r < 1$. Let $\hat{\mathbb{E}}[Y|X]$ be the empirical conditional expectation of $Y$ given $X$ based upon $m$ i.i.d. samples of $(X, Y)$ and define $h(X) := \min(\max(\mathbb{E}[Y|X], r), -r)$. Then with probability at least $1 - \delta$,

$$\mathbb{E}[(\tanh^{-1}(h(X)) - \tanh^{-1}(\mathbb{E}[Y|X]))^2] \lesssim \frac{2^n}{\gamma(1 - r^2)^2 m} + \frac{\log(1/\delta)}{(1 - r^2)^2 m}$$

We will prove the result by proving the analogous result without the $\tanh^{-1}$ first, as Lemma 14. The following general result reduces this to computing the local Rademacher complexity of the corresponding function class.

**Theorem 14** (Corollary 5.3 of [60])**.** *Suppose that $\mathcal{F}$ is a class of functions from $\mathcal{X}$ to $[-1, 1]$ and $\ell(\hat{y}, y)$ is a loss which satisfies:*

1. *$\ell$ is L-Lipschitz in $\hat{y}$.*

2. *There is a constant $B \geq 1$ such that for any random variable $X$ supported on $\mathcal{X}$ and random variable $Y$ on $[-1, 1]$*

$$\mathbb{E}(f(X) - f^*(X))^2 \leq B\mathbb{E}[\ell(f(X), Y) - \ell(f^*(X), Y)]$$

   *where $f^*(X)$ is a minimizer of $\mathbb{E}[\ell(f(X), Y)]$ which we assume exists.*

*Then if $\psi(r)$ is a sub-root function (meaning a monotonically increasing non-negative function with $\psi(r)/\sqrt{r}$ monotonically decreasing) such that*

$$\psi(r) \geq BL\mathbb{E} \sup_{f \in \mathcal{F}, L^2\mathbb{E}[(f - f^*)^2] \leq r} \frac{1}{m} \sum_{i=1}^{m} \sigma_i(f - f^*)(X_i) \tag{5}$$

*where the $\sigma_i$ are i.i.d. Rademacher random variables, then for any $r \geq \psi(r)$ with probability at least $1 - \delta$*

$$\mathbb{E}[\ell(\hat{f}(X), Y) - \ell(f^*(X), Y)] \lesssim \frac{r}{B} + \frac{(L + B)\log(1/\delta)}{m}$$

*where the notation $\lesssim$ hides an absolute constant.*

**Lemma 14.** *Under the same setup as Lemma 12,*

$$\mathbb{E}[(h(X) - \mathbb{E}[Y|X])^2] \lesssim \frac{2^n}{\gamma m} + \frac{\log(1/\delta)}{m}.$$

*Proof.* We consider $\mathcal{F}$ the class of arbitrary functions from $\mathcal{X}$ to $[-r, r]$ and take $\ell(\hat{y}, y) := (\hat{y} - y)^2$ to be the square loss so $L = 2$ and $B = 1$ satisfy the conditions above. It is clear from the definition of $h$ that it is the empirical risk minimizer for this function class and loss. Since this class is convex we can take $\psi(r)$ to be defined by the rhs of (5) (Lemma 3.4 of [60]) and it remains to compute the fixed point of $\psi$. Thus if we write $g := f - f^*$

$$\psi(r) = 2\mathbb{E} \sup_{f:4\mathbb{E}[g^2] \leq r} \frac{1}{m} \sum_{i=1}^{m} \sigma_i g(X_i)$$

and we observe by the assumption $\Pr(X = x) \geq \gamma/2^n$ that

$$\mathbb{E}_X[g^2] \geq \gamma \mathbb{E}_{X' \sim Uni(\{\pm 1\}^n)}[g(X')^2] = \gamma \sum_S \widehat{g}(S)^2$$

by Plancherel's Theorem [52] where $\widehat{g}(S)$ denotes the Fourier coefficient of $g$ corresponding to set $S$, so that $g(x) = \sum_S \widehat{g}(S)x_S$ where $x_S = \prod_{s \in S} x_s$. Therefore by the above, the Cauchy-Schwarz inequality, and Jensen's inequality we have

$$\psi(r) = 2\mathbb{E} \sup_{g:4\mathbb{E}[g^2] \leq r} \frac{1}{m} \sum_{i=1}^{m} \sigma_i g(X_i)$$

$$\leq 2\mathbb{E} \sup_{g:\sum_S \hat{g}(S)^2 \leq r/4\gamma} \frac{1}{m} \sum_S \hat{g}(S) \frac{1}{m} \sum_{i=1}^{m} \sigma_i(X_i)_S$$

$$\leq \sqrt{r/\gamma} \mathbb{E} \frac{1}{m} \sqrt{\sum_S \left( \sum_{i=1}^{m} \sigma_i(X_i)_S \right)^2}$$

$$\leq \frac{\sqrt{r}}{m\sqrt{\gamma}} \sqrt{\mathbb{E} \sum_S \left( \sum_{i=1}^{m} \sigma_i(X_i)_S \right)^2} = \frac{\sqrt{r}}{\sqrt{m\gamma}} 2^{n/2}.$$

Solving for the fixed point of $r = \frac{\sqrt{r}}{\sqrt{m\gamma}} 2^{n/2}$ gives $r^* = \frac{2^n}{\gamma m}$ so the result follows from Theorem 14. $\square$

*Proof of Lemma 12.* Recall that the derivative of $\tanh^{-1}$ at $x$ is $\frac{1}{1-x^2}$. Therefore on the domain $[-r, r]$ the function $\tanh^{-1}$ is $\frac{1}{1-r^2}$ Lipschitz. Therefore by the mean value theorem,

$$\mathbb{E}[(\tanh^{-1}(h(X)) - \tanh^{-1}(\mathbb{E}[Y|X]))^2] \leq \frac{1}{(1-r^2)^2}\mathbb{E}[(h(X) - \mathbb{E}[Y|X])^2]$$

and applying Lemma 14 gives the result. $\qquad\square$

### D.3 Matching Computational Lower Bounds

In the following sequence of theorems we show that our runtime guarantees for structure learning of RBMs cannot be significantly improved. The first result relies in part on the representation of sparse parity with noise given in [18]; this embedding is constructed in a similar way to the first embedding used in Theorem 11. It shows the dependence on $\lambda_1$ and $\eta$ is correct when asking for structure recovery.

**Theorem 15.** *In the same setup as Theorem 12 and under Assumption 1, there exists a family of instances parameterized by $n$ going to infinity with $\lambda_2 \leq 2$ such that any algorithm which is able to achieve structure recovery for a model with all neighbors being $\eta$-nondegenerate requires runtime $n^{\Omega(\log(\lambda_1/\eta)/\log\log(\lambda_1/\eta))}$, regardless of its sample complexity.*

*Proof.* In [18], it was shown that for any fixed constant $\eta$ (say $\eta = 1/8$), there exists an embedding of $k$-sparse parity with noise into an RBM where every hidden unit has incoming edges of total $\ell_1$ norm upper bounded by 2 (i.e. satisfying $\lambda_1 \leq 2$) and there are $2^{O(k)}$ hidden units; it can be checked straightforwardly that for $\eta = 1/8$ that $\lambda_2 = k^{O(k)}$. Therefore if we fix $\epsilon = \eta/2$ then when assuming the hardness of $k$-sparse parity with noise there is a $n^{\Omega(k)}$ runtime lower bound which matches since $\lambda_2 = e^{O(k)}$.

For the tightness in $\epsilon$, by making the parity bias $\eta$ exponentially small in $k\log(k)$, it's easy to check that by repeating the construction in [18] that we can make $\lambda_2$ a constant; then to find the parity with noise one needs $\epsilon$ exponentially small in $k\log k$ as well, and the hardness assumption implies the runtime must be $n^{\Omega(k)}$. $\qquad\square$

By tensorizing this construction, we show that the $\eta$-nondegeneracy assumption is required, even if we only care about distribution learning. More precisely, we need it to learn in TV distance with runtime better than the pessimistic $n^{O(d_h)}$ result which follows from viewing the RBM as an unstructured MRF and using the result of [8].

**Theorem 16.** *There exists a family of RBMs with $n$ nodes, maximum hidden node degree $d_H$, and $\lambda_1, \lambda_2 = O(1)$ such that any algorithm which can learn this family of RBMs within total variation distance at most $1/4$ requires $n^{\Omega(d_H)}$ time.*

*Proof.* The construction in Theorem 15 shows that there exists a family of RBMs given by embedding sparse parity with noise with the desired property, except that the total variation distance is only guaranteed to be $2^{-O(d_H\log(d_H))}$. By building a larger RBM consisting of $2^{d_H\log(d_H)}$ disjoint copies of the original RBM (note that the resulting increase in $n$ is a multiplicative factor independent of the original $n$), we can boost the total variation distance to be arbitrarily close to 1. $\qquad\square$

In order to give lower bounds with respect to $\lambda_2$ for fixed $\eta$, we need a significantly more involved argument. We first recall an approximate construction of parity (with low levels of noise) from [48]:

**Theorem 17** (Theorem 7 of [48])**.** *There exists an RBM network with $n^2 + 1$ hidden units and weights $poly(n, \log(1/\epsilon))$ such that the marginal distribution $P$ on the visible units satisfies $P(x) \propto e^{f(x)}$ for some $f$ satisfying*

$$\sup_{x \in \{\pm 1\}^n} |f(x)/C - x_1 \cdots x_n| \leq \epsilon$$

*where $C > 0$ satisfies $C = poly(\log(n), \log(1/\epsilon))$.*

This construction is for a dense parity, but obviously we can make the parity as sparse as we want by adding additional visible units not connected to anything else. More significantly, since the above

theorem only constructs an $\epsilon$-approximate instance of parity with noise $\eta = O(1/2 - 1/poly(n, 1/\epsilon))$, when $n$ or $1/\epsilon$ is large it does not seem that the resulting distribution is computationally hard to distinguish from the uniform distribution, since Gaussian elimination over $\mathbb{F}_2$ has some chance of succeeding to find the parity. Since we need $\epsilon$ to be small for the model to be indistinguishable from sparse parity with noise, this appears to be a barrier to deriving a hardness result from the above Theorem. Instead, we will prove that our result cannot be significantly improved for SQ (Statistical Query) algorithms (for a reference, see [49]). In the Statistical Query model algorithms do not have access to data, but instead have access to an SQ oracle:

**Definition 6.** *An oracle for the statistical query model over distribution $\mathcal{D}$ over $X, Y$ takes input $(g, \tau)$ where $g$ is a function $g : \{\pm 1\}^n \times \{\pm 1\} \to [-1, 1]$ and $\tau$ is a tolerance, and gives output $v$ with*

$$|\mathbb{E}_{X,Y \sim D}[g(X, Y)] - v| \leq \tau.$$

Standard arguments, i.e. implementing the needed regressions using standard gradient-based methods for convex optimization shows that our algorithm for learning RBMs can be implemented in the statistical query model (in this case, the separation of $X$ and $Y$ in the definition above is somewhat artificial but we will take $Y$ to be a particular visible unit in the RBM). We will show that statistical query algorithms cannot do better than subexponential dependence on $\lambda_2$.

The following theorem statements a lower bound for learning concepts of large SQ-dimension in the Statistical Query model. The definition of SQ-dimension can be found in [49], but for our purposes the only needed fact is that the class of $k$-parities over the uniform distribution $\{\pm 1\}^n$ has SQ-dimension $\binom{n}{k}$ [49].

**Theorem 18** ([49]). *Let $\mathcal{F}$ be a class of functions over $\{\pm 1\}^n$ and $D$ a distribution such that $SQ\text{-}DIM(\mathcal{F}, D) \geq d \geq 16$. Then if all queries are made with tolerance at least $1/d^{1/3}$, then at least $d^{1/3}/2$ queries are required to learn $\mathcal{F}$ with error less than $1/2 - 1/d^3$ in the statistical query model.*

**Theorem 19.** *Let $S$ be an unknown subset of $[n]$ of size $k$ and containing $n$ and $\mathcal{D}$ is the distribution of the RBM produced by Theorem 17 on $S$ where the other $n - |S|$ visible units are isolated and without external field. Let $\mathcal{F}$ be the class of parities on $[n-1]$. As before, $\lambda_2$ refers to the maximum $\ell_1$-norm into any hidden unit and we choose parameters so that $\lambda_2 = poly(n)$ and $\|w\|_1 = poly(n)$. There exists $\epsilon > 0$ so that no SQ algorithm with tolerance $n^{-\lambda_2^\epsilon}$ and access to $n^{\lambda_2^\epsilon}$ queries can learn $\mathcal{F}$ with error less than $1/4$.*

*Proof.* In Theorem 17 we take $\epsilon = \exp(-n)$ which gives $\lambda_2 = poly(n)$. The resulting RBM is then within TV distance $\exp(-n)$ of the distribution of a parity over the uniform distribution with a small amount of label noise, so an SQ algorithm for the RBM setting implies an SQ algorithm for learning parity, and the result follows from the lower bound of Theorem 18. □

# E  Learning a Feedforward Network by Learning RBMs

In this section we reverse the connection between RBMs and Feedforward networks by using RBMs with certain structural assumptions as a useful *distributional assumption* for learning feedforward network. More formally, we assume our data is generated by the following Supervised RBM.

**Definition 7.** *A* Supervised Restricted Boltzmann Machine *is any joint distribution over random variables $X$ valued in $\{\pm 1\}^{n_1}$, $H$ valued in $\{\pm 1\}^{n_2}$ and label $Y \in \{\pm 1\}$ of the form*

$$\Pr[X = x, H = h, Y = y] \propto \exp\left(\langle x, Wh \rangle + \langle h, w \rangle y + \langle b^{(1)}, x \rangle + \langle b^{(2)}, h \rangle + b^{(3)} y\right)$$

*where the* weight matrix $W$ *is an arbitrary $n_V \times n_H$ matrix and* external fields/biases $b^{(1)} \in \mathbb{R}^{n_1}$, $b^{(2)} \in \mathbb{R}^{n_2}$ and $b^{(3)}$ are arbitrary, and $X$ is referred to as the vector of visible unit activations and $H$ the vector of hidden unit activations.*

We make the following additional assumptions on the parameters of the model.

**Assumption 2** (Minimum Ferromagnetic Interaction). *For all $i \in [n_1], j \in [n_2]$ either $W_{ij} = 0$ or $W_{ij} \geq \alpha$.*

We do not make any assumption on the weight $w$ to the label. Therefore the model overall is not ferromagnetic.

**Assumption 3** (Sparsity). *For all $i \in [n_1]$, $\sum_{j=1}^{n_2} W_{ij} + |b_i^{(1)}| \le \lambda$ and for either $y = -1$ or $y = 1$, for all $j \in [n_2]$ $\sum_{i=1}^{n_1} W_{ij} + |b_j^{(2)} + yw_j| \le \lambda$.*

Here the sparsity assumption implies that under the conditioning of the label to either value, the sparsity parameter is bounded. This conditional sparsity can be exploited by an algorithm for learning the conditional distribution whereas a direct regression algorithm may be unable to gain from the same.

**Remark 7.** *Observe that the generative model of $X$ itself is not sparse since $Y$ is connected to all hidden nodes however conditioned on knowing the label $Y$, the model is now sparse. This assumption is more reasonable than assuming sparsity directly on the model of $X$ which may not hold.*

**Assumption 4** (Balanced Label). *For $y \in \{\pm 1\}$, $\Pr[Y = y] \ge \beta$.*

The above assumption essentially rules out trivial constant learners. Using data, it is easy to check if this assumption is satisfied or not.

As before, we can compute the conditional mean function of the label as follows:

$$\mathbb{E}[Y|X = x] = \tanh\left(b^{(3)} + \sum_j \tanh^{-1}\left(\tanh(w_j)\nu_j\right)\right)$$

where $\nu_j := \tanh\left(b_j^{(2)} + \sum_i \tanh^{-1}\left(\tanh(W_{ij})X_i\right)\right) = \tanh\left(b_j^{(2)} + \sum_i W_{ij}X_i\right)$. This represents a 2-layer neural network and in the limit of infinite hidden nodes, it can represent all 2-layer $\tanh$ networks (see Lemma 2).

**Assumption 5** (Boundedness). *When $\mathbb{E}[Y|X = x]$ is re-expressed as $\tanh(f^*(x) + b^*)$ for some function $f^*$ with no constant term and $b^* \in \mathbb{R}$. $|b^*| \le B$ for some $B > 0$.*

The above assumption intuitively says that the effect on $Y$ that does not depend on $X$ is bounded. $B$ can be bounded in terms of the network parameters.

Also observe that conditioned on a fixed label,

$$\Pr[X = x, H = h|Y = y] \propto \exp\left(\langle x, Wh \rangle + \langle b^{(1)}, x \rangle + \langle b^{(2)} + wy, h \rangle\right)$$

which is a sparse, ferromagnetic RBM with arbitrary external field. Thus, we capture a neural network problem with a conditional RBM distributional assumption on the input. This distributional assumption seems more natural than the Gaussian input distribution which is extensively used in prior work. Also, this assumption allows us to leverage prior known algorithms for structure learning of ferromagnetic RBMs to learn the prediction function.

### E.1 Preliminaries: Structure Learning of RBMs with Ferromagnetic Interactions

Consider a RBM with the following additional assumptions:

**Assumption 6** (Minimum Ferromagnetic Interaction). *For all $i \in [n_1], j \in [n_2]$ either $W_{ij} = 0$ or $W_{ij} \ge \alpha$.*

**Assumption 7** (Sparsity). *For all $i \in [n_1]$, $\sum_{j=1}^{n_2} W_{ij} + |b_i^{(1)}| \le \lambda$ and for all $j \in [n_2]$, $\sum_{i=1}^{n_1} W_{ij} + |b_j^{(2)}| \le \lambda$.*

Under these assumptions, [19] has shown that a simple greedy algorithm based on covariance maximization suffices to learn the structure of the RBM. Under the further assumption of non-negative external fields, [18] previously showed a similar greedy maximization algorithm with better dependence on the sparsity parameter $\lambda$.

The crucial structural property that [19] use is their algorithm is the following strengthening of the FKG inequality,

**Lemma 15** (Lemma 2 of [19]). *For any observed nodes $u, v$ and set $S \subseteq [n_1]\backslash\{u, v\}$,*

$$\mathsf{Cov}(u, v|X_S = x_S) := \mathbb{E}[X_u X_v|X_S = x_S] - \mathbb{E}[X_u|X_S = x_S]\,\mathbb{E}[X_v|X_S = x_S] \ge \alpha^2 \exp(-12\lambda).$$

Subsequently they define *average conditional covariance* $\mathsf{Cov}^{\mathsf{Avg}}(u, v|S) = \mathbb{E}_{x_S}[\mathsf{Cov}(u, v|X_S = x_S)]$ which straightforwardly is lower bounded by an application of the above lemma. Their final algorithm essentially greedily maximizes this average conditional covariance to build the neighborhood.

**Theorem 20** (Theorem 2 of [19])**.** *Consider $M$ samples $\mathcal{S}$ drawn from a RBM with arbitrary external field satisfying the given assumptions. For $\tau = \frac{\alpha^2}{2} \exp(-12\lambda)$ and $\delta = \exp(-2\lambda)/2$, with probability $1 - \zeta$, LEARNRBMNBHD$(u, \tau, \mathcal{S})$ outputs* exactly *the two-hop neighborhood of observed variable $u$ for*

$$M \geq \Omega\left((\log(1/\zeta) + T^* \log(n)) \frac{2^{2T^*}}{\tau^2 \delta^{2T^*}}\right) \text{ and } T^* = \frac{8}{\tau^2}.$$

*Moreover, the algorithm runs in time $O(T^*Mn)$.*

## E.2 Prediction from Distribution Learning

Here we will present our algorithm for learning the supervised RBM followed by a proof of its correctness. Instead of learning the label function directly, we will instead first learn the underlying generative model of $X$ conditioned on a particular value of the label and use this knowledge to predict $Y$.

**Theorem 21.** *Given a supervised RBM satisfying Assumption 2, 3, 4 and 5, there exists an algorithm with sample complexity $m = n^2 \exp(\lambda)^{\exp(O(\lambda))}(1/\alpha)^{O(1)}(1/\beta)^{O(1)} \log(n/\delta)/\epsilon^2$ and runtime $poly(m)$ returns hypothesis $h$ such that,*

$$\mathbb{E}[\ell(h(X), Y] - \mathbb{E}[\ell(h^*(X), Y] \leq \epsilon$$

*where $\ell$ is the logistic loss and $h^*$ is the minimizer of the logistic loss.*

**Remark 8.** *For an example where this algorithm is better than if we have no distributional assumptions, observe that we can construct a ferromagnetic RBM where $\mathbb{E}[Y|X]$ is a sparse parity function by adapting in a straightforward way the reduction used in the proof of the part of Theorem 11 with bounded $\lambda$ (the use of $\tanh$ as opposed to $f_\beta$ in that construction is not fundamental, or we can use a finite version of Lemma 2), since the hidden units in that proof all have nonnegative weights. It's clear why Algorithm LEARNSUPERVISEDRBMNBHD is better than an algorithm which doesn't know the input distribution: under the true input distribution, the visible units involved in the parity are correlated so the algorithm can find them, which makes learning the sparse parity easy.*

Our main algorithm can be broken down into three main steps: 1) Use greedy maximization (similar to Algorithm 1 of [19]) to first learn the two-hop neighborhood $\mathcal{N}(i)$ of each observed variable $i$ w.r.t. the hidden layer conditioned on the label, 2) For each observed variable $X_i$, learn the distribution for $X|Y = y$ for $y = \pm 1$, and 3) Use the estimated distribution to compute $\mathbb{E}[Y|X]$.

**Structure Learning** For notation simplicity, we will overload notation and represent $\mathsf{Cov}^{\mathsf{Avg}}(u, v|S, Y) = \mathbb{E}_{x_S,y}[\mathsf{Cov}(u, v|X_S = x_S, Y = y)]$ where $\mathsf{Cov}(u, v|X_S = x_S, Y = y) = \mathbb{E}[X_u X_v|X_S = x_S, Y = y] - \mathbb{E}[X_u|X_S = x_S, Y = y]\,\mathbb{E}[X_v|X_S = x_S, Y = y]$. Then for structure learning, our algorithm essentially follows Algorithm 1 of [19] with the slight modification of conditioning w.r.t. $Y$.

**Theorem 22.** *Consider $m$ samples $\mathcal{S}$ drawn from a supervised RBM satisfying Assumption 2, 3 and 4. For $\tau = \frac{\beta\alpha^2}{2} \exp(-12\lambda)$ and $\delta = \exp(-2\lambda)/2$, with probability $1 - \zeta$, LEARNSUPERVISEDRBMNBHD$(u, \tau, \mathcal{S})$ outputs* exactly *the two-hop neighbors of observed variable $u$ w.r.t. the hidden layer, with*

$$m \geq \Omega\left((\log(1/\zeta) + T^* \log(n)) \frac{2^{2T^*}}{\tau^2 \beta \delta^{2T^*}}\right) \text{ and } T^* = \frac{8}{\tau^2}.$$

*Moreover, the algorithm runs in time $O(T^*Mn)$.*

*Proof.* In order to apply Theorem 20 to our setting, the only two properties we need to show are 1) given the conditioning of $Y$, the average conditional covariance bound still holds, that is,

$\mathsf{Cov}^{\mathsf{Avg}}(u,v|S\cup\{0\})$ is lower bounded for all $S\subseteq[n_2]\backslash\{u,v\}$ for $v$ in the two-hop neighborhood of $u$, 2) $\Pr[X_S=x_S,Y=y]$ for all $x_S$ and $y$. We have,

$$\mathsf{Cov}^{\mathsf{Avg}}(u,v|S,Y)=\sum_{y\in\pm1}\sum_{x_S\in\{\pm1\}^{|S|}}\Pr[X_S=x_S,Y=y]\mathsf{Cov}(u,v|X_S=x_S,Y=y)$$

By Assumption 3, we know that either for $y=1$ or $y=-1$ (say $y=1$ WLOG), the resulting RBM is sparse therefore we can apply Lemma 15 to the ones conditioned on $y=1$. Also, we know that $\mathsf{Cov}(u,v|X_S=x_S,Y=y)\geq0$ for all $x_S$ and $y$ due to FKG inequality for ferromagnetic RBMs. This implies that,

$$\mathsf{Cov}^{\mathsf{Avg}}(u,v|S,Y)\geq\sum_{x_S\in\{\pm1\}^{|S|}}\Pr[X_S=x_S,Y=1]\mathsf{Cov}(u,v|X_S=x_S,Y=1)$$

$$\geq\sum_{x_S\in\{\pm1\}^{|S|}}\Pr[X_S=x_S,Y=1]\alpha^2\exp(-12\lambda)$$

$$\geq\Pr[Y=1]\alpha^2\exp(-12\lambda)\geq\beta\alpha^2\exp(-12\lambda).$$

For the second part, let us order the elements of $S$ of size $k$ as $s_1,\ldots,s_k$, then we have

$$\Pr[X_S=x_S,Y=y]=\Pr[Y=y]\times\Pr[X_{s_1}=x_{s_1}|Y=y]\times\Pr[X_{s_2}=x_{s_2}|X_{s_1}=x_{s_1},Y=y]\times\ldots$$
$$\times\Pr[X_{s_k}=x_{s_k}|X_{s_1}=x_{s_1},\ldots,X_{s_{k-1}}=x_{s_{k-1}},Y=y]$$

Since $l_1$-norm to the observed nodes is bounded by $\lambda$, by Bresler's property (see [6]) we have $\Pr[X_{s_r}=x_{s_r}|X_{s_1}=x_{s_1},\ldots,X_{s_r}=x_{s_r},Y=y]\geq\delta$. This implies that $\Pr[X_S=x_S,Y=y]\geq\beta\delta^{|S|}$ for all values of $x_S$ and $y$. Now by applying Theorem 20 with the correct parameters, we get the required result. $\qquad\square$

**Distribution Learning**   Given the neighborhood of each observed node, we run Algorithm DISTRI-BUTIONFROMSTRUCTURE and subsequently use Lemma 13 to guarantee that we obtin the weights of the unnormalized MRFs for distributions $X|Y=y$ for $y\in\{\pm1\}$ up to epsilon accuracy. More formally,

**Lemma 16.** *Let the maximum two-hop degree of any visible node is at most $d_2$ and $\|b^{(1)}\|_\infty\leq B$. For $\delta>0$ and $m=\Omega\left(n^2\left(\frac{2}{(1-\tanh(\lambda))}\right)^{d_2+1}\log(n/\delta)/\epsilon^2\right)$ we have that with probability at least $1-\delta$, Algorithm DISTRIBUTIONFROMSTRUCTURE given $m$ samples and $\widehat{\mathcal{N}}(i)=\mathcal{N}(i)$ for every $i$ returns unnormalized MRFs of $X|Y=y$ for $y\in\{\pm1\}$ with coefficients $\hat{f}_S^{(y)}$ that are close to the coefficients of the true unnormalized MRFs $f_S^{(y)}$, that is,*

$$\sum_S|\hat{f}_S^{(y)}-f_S^{(y)}|\leq\epsilon.$$

**Constructing the Predictor**   Observe that the joint distribution of $X$ and $Y$ can be represented as,

$$\Pr[X=x,Y=y]\propto\exp\left(\sum_Sf_S^{(1)}x_S\mathbb{1}[y=1]+\sum_Sf_S^{(-1)}x_S\mathbb{1}[y=-1]+b^*y\right)$$

for some $b^*$ and coefficients of the true unnormalized MRFs $f_S^{(y)}$ corresponding to conditioning of $Y=y$. This gives us,

$$\mathbb{E}[Y|X=x]=\tanh\left(\sum_S\frac{(f_S^{(1)}-f_S^{(-1)})}{2}x_S+b\right)\approx_\epsilon\tanh\left(\sum_S\frac{(\hat{f}_S^{(1)}-\hat{f}_S^{(-1)})}{2}x_S+b\right)$$

Since we have estimates of $f_S^{(y)}$, to learn the predictor for $Y$ we only need to find $b^*$ which we can find by minimizing $\ell$ snce it is convex. Let $h_b=\sum_S\frac{(f_S^{(1)}-f_S^{(-1)})}{2}x_S+b$ and $\hat{h}_b=\sum_S\frac{(\hat{f}_S^{(1)}-\hat{f}_S^{(-1)})}{2}x_S+b$. We minimize $\hat{E}[\ell(h_b(X),Y)]$ over $b$ and suppose the minimizer is $\hat{b}$. By Fact 1.3, $\ell(\hat{h}_b(X),Y)\leq\ell(h_b(X),Y)+4\epsilon$. By Fact 1.4, $h_{b^*}$ is the minimizer of the logistic loss. Then we have,

$$\hat{\mathbb{E}}[\ell(h_b(X),Y)]\leq\hat{\mathbb{E}}[\ell(\hat{h}_{b^*}(X),Y)]+4\epsilon\leq\hat{\mathbb{E}}[\ell(h_{b^*}(X),Y)]+8\epsilon.$$

Last we need a generalization bound that holds for our hypothesis class. For this we bound the Rademacher complexity (see [56] for more background) of the class of functions $\ell \circ \mathcal{H}$ where $\mathcal{H} := \{h_b | |b| \leq B\}$.

$$\mathcal{R}_m(\ell \circ \mathcal{H}) \leq 2\mathcal{R}_m(\mathcal{H})$$

$$= \mathbb{E}_\sigma \left[ \sum_{b||b| \leq B} \frac{1}{m} \sum_{i=1}^m \sigma_i h_b(x^{(i)}) \right]$$

$$= \mathbb{E}_\sigma \left[ \sum_{b||b| \leq B} \frac{1}{m} \sum_{i=1}^m \sigma_i \sum_S (f_S^{(1)} - f_S^{(-1)}) x_S + 2b \right]$$

$$= 2\mathbb{E}_\sigma \left[ \sum_{b||b| \leq B} \frac{1}{m} \sum_{i=1}^m \sigma_i b \right]$$

$$= 2B\mathbb{E}_\sigma \left[ \frac{1}{m} \left| \sum_{i=1}^m \sigma_i \right| \right]$$

$$\leq \frac{2B}{\sqrt{m}}.$$

Here the first inequality follows from the contraction lemma (see [61]) and the last from standard properties of Radmeacher variables. Now applying Theorem 26.5 from [56] we get

$$|\mathbb{E}[\ell(h_b(X), Y)] - \mathbb{E}[\ell(\hat{h}_b(X), Y)]| \leq \frac{2B}{\sqrt{m}} + c\sqrt{\frac{\log(1/\delta)}{\sqrt{m}}}$$

where $c$ is the maximum value of logistic loss by any hypothesis in the class. Observe that by Fact 1.4, logistic loss at $h_{b^*}$ is bounded by a constant. Hence by Lipschitzness, we know that loss anywhere will be bounded by $O(\max(1, B))$. Therefore choosing $m \geq \Omega(B^2 \log(1/\delta)/\epsilon^2)$ suffices to get within $\epsilon$. Combining this with before we get that the loss is within $O(\varepsilon)$ of the best loss.

**Proof of Theorem 21**    First, the algorithm runs LEARNSUPERVISEDRBMMBHD for each node to learn the structure of the induced RBM exactly with the given samples

$$m_1 = \exp(\lambda)^{\exp(O(\lambda))} (1/\alpha)^{O(1)} (1/\beta)^{O(1)} \log(n/\delta).$$

With the structure, we run DISTRIBUTIONFROMSTRUCTURE to learn both the induced RBMs for each conditioning of the label using $m_2 \geq \Omega \left( n^2 \left( \frac{2}{(1-\tanh(\lambda))} \right)^{d_2+1} \log(n/\delta)/\epsilon^2 \right)$ samples where $d_2$ is the max 2-hop neighborhood size. Note that the dependence on $\lambda$ is greater in $m_1$ than $m_2$. Subsequently, given the unnormalized mrfs, we run a simple optimization to find the bias term of the predictor using $m_3 \geq \Omega(B^2 \log(1/\delta)/\epsilon^2)$ samples. Combining the learnt mrf and the bias term, we get our hypothesis.

**Remark 9.** *If the model is not ferromagnetic, it is also possible and we expect it may be advantageous in some models to still use a similar indirect approach based on Bayes rule for learning a predictor of $Y$, but using the result of Theorem 1 instead of the greedy structure recovery method used in this section. The disadvantage of this approach is of course that its runtime for achieving structure recovery is slower.*

# F    Additional Experimental Data

Figure 3 contains samples generated from the model trained on MNIST images. For reference, we also include samples from the true MNIST and FashionMNIST training sets in the same format as Figure 2 and Figure 1.

Figure 2: Five i.i.d. samples for each MNIST class, drawn from the trained model by Gibbs sampling.

Figure 3: Reference MNIST images chosen randomly from training set.

Figure 4: Reference FashionMNIST samples from training set.

[Supplementary Material 2]