[Reviews · NeurIPS 2020]

Review 1

Summary and Contributions: The paper discusses algorithmic guarantees for the learning of RBM models according to sample size and some measure of the complexity of the RBM models. The paper discusses both the unsupervised learning and the supervised learning case, for the latter an algorithm for learning an RBM is also proposed.

Strengths: The paper adresses the problem of learning a powerful class of generative models improving on previous works which required stronger assumptions on the parameters of the RBM being learned. The paper explains clearly its reasoning: e.g. drawing parallels with two layer neural networks and conditional expectations in RBMs, or explaining the need for a complexity measure to characterize the hardness of the learning from a RBM. An algorithm derived from the theory to learn a RBM encoding a classification task is furthermore tested on a real dataset and shown to achieve comparable performance to algorithms with related levels of complexity.

Weaknesses: The main text is very dense given the page limit and some informal statements are not entirely clear (see questions below).

Correctness: Given the allowed time, I could not check the proofs. The method for the proposed experiment seem correct.

Clarity: Overall the paper is well written. The main text is quite dense but it is unclear that this could be improved given the page constraint. There are a few places where some variables lack definitions (see details below).

Relation to Prior Work: The paper clearly explains the context of the contribution in the introduction and further details relation to prior works in a dedicated section.

Reproducibility: Yes

Additional Feedback: Definitions: - Line 130-131 theorem 2, the loss function \ell (logistic?) is not defined in the main text. - In Example 1 and 2 page 5 and line 218, what is the definition of \eta? The maximum \eta such that the RBM has all of its pairs \eta-degenerate? This is not entirely clear. - Page 5 line 211 - Is the parameter d defined ? References: Page 2 line 57 - “a natural generalization of fully-observed Ising models, for which the learning problem is well understood.” Could the authors provide corresponding references? Typos: - Abstract line 4 - the most well studied - the best studied - Page 2 line 38 - “statistical” <- statistically - Page 4 line 159 - “the each” - Page 4 line 164, “(e.” ***** I have read other reviews and authors feedback, which confirms my positive assessment of the paper. The authors answered to the clarification I was requesting. I am hoping they will take advantage of the additional page to make sure they implement these clarifications in the text.


Review 2

Summary and Contributions: This paper proves several learning guarantees for two-layer feedforward neural networks and for Restricted Bolzmann Machines, gives nearly matching lower bounds (under hardness assumptions), and explores connections between the two settings in a rigorous manner. Specifically, the authors give an algorithm for structure learning of RBMs under a reasonable-seeming assumption generalizing ferromagneticity. This is proved via a new guarantee for learning single-hidden-layer neural networks under L^infty bounded inputs. Finally, the authors give guarantees for a "supervised RBM" classification algorithm, where the label is viewed as one of the visible units in an RBM, under assumptions about the weights in this RBM.

Strengths: This paper gives new algorithms with strong guarantees for basic neural network models. By exploring connections between single hidden layer feedforward networks and RBMs, the authors introduce new analytic techniques in this setting. This is a solid contribution.

Weaknesses: Theorem 6 doesn't seem as well motivated. Its assumptions seem strong and very specialized, and the experimental implementation is (unsurprisingly) lackluster in its performance. And we already have lots of guarantees for training single-hidden layer NNs, although this paper gives better bounds under some assumptions.

Correctness: I have no concerns about correctness

Clarity: The writing is excellent. I would suggest dropping the poorly motivated experiments section to make room for the introduction section of the appendices, which gives a clearer roadmap of the proofs.

Relation to Prior Work: The authors do a good job of putting Theorem 2 in context. I am less familiar with the RBM literature, and the authors don't seem to have done as much to give context to Theorem 4. More discussion there would be appreciated.

Reproducibility: Yes

Additional Feedback: I am grateful to the authors for their response. I hope they include some additional context for Theorem 4 in the final version.


Review 3

Summary and Contributions: The paper considers learning Restricted Boltzmann Machines (RBMs). The assumption is that the RBM has bounded normed weights, and non-degenerated correlations between any two-hop neighboring visible units. This is different from previous works that either assume ferromagneticity or bounded max hidden degree. The goal is structure learning (that is, recovering the Markov Blanket of the visible units, which can then imply say distributional learning). The analysis is done via a connection to feedforward networks: predicting one visible unit from the others can be done via a two-layer feedforward network. It then gives a learning method to learn those networks (with nearly optimal sample and runtime complexity under the conjectured hardness of sparse parity with noise). Using the connection, the paper also provides the theoretical study of supervised RBMs, giving an algorithm for learning a natural class of supervised RBMs, and gives empirical supports.

Strengths: + The topic is interesting. RBMs are probably the most well-known neural/graphical models for unsupervised learning. Theoretical analysis will provide better insights into this learning architecture. + The connection from RBMs to feedforward neural networks is interesting, though it is not very complicated. This will allow borrowing tools from the latter area to the former, as did in this paper.

Weaknesses: - The learning method for the feedforward networks is by l1-regularized regression over monomials. This is not connected with popular practical algorithms for RBMs. Can the authors comment on any insights gained for the practical algorithms from the connection between RBMs and the feedforward networks? ============================after response==================== The authors has answered my quedstion on relation/implications to practical algorithms for training RBMs. It doesn't completely address my concerns, but it gives good reasons for the study in the paper, so I increased the scores to 7.

Correctness: I think the theoretical part is correct, though I only verify the key lemmas.

Clarity: Yes. But there seem to be some typos, eg, -- Line 100: what are n_V and n_H? -- Theorem 2: d should be D? -- X_{~i} should be X_{~i,j} in the math display in Definition 1

Relation to Prior Work: Clear.

Reproducibility: Yes

Additional Feedback:


Review 4

Summary and Contributions: Post Rebuttal: Thanks for clarifications. I am changing the score to 7. The paper gives a new algorithm for learning the structure Restricted Boltzmann Machines (formalized using Markov blankets), which is claimed to work for larger parameter regimes than the previous work. This is done by considering the problem of predicting the spin of a node given the spins of all other nodes. This dependence is shown to be given by a one-hidden layer neural net (with somewhat non-standard activations). An algorithm for learning this network is given based on polynomial approximation of the neural net and using regression on degree-D monomial feature map (with \ell_1 constraint). The algorithm works under L_\inf constraint on the input vector which is different from the past work. Given the above algorithm for learning the dependence of one node on the rest, under suitable non-degeneracy conditions, an algorithm is given for learning the structure (Markov blanket) of the RBM. Nearly matching lower bounds are provided (under hardness assumptions or in the SQ model). The reduction to neural networks is also used for learning supervised RBMs, which can be thought of as a neural network under distributional assumptions on the data (in terms of "sparsity and nonnegative correlations among the input features 307 conditional on the output label"). This distributional assumptions seems to be new.

Strengths: The algorithm is conceptually simple, and in terms of the chosen complexity parameters (in terms of \lambda_1, \lambda_2) the performance guarantees are nearly tight.

Weaknesses: The main result is claimed to "often" expand the parameter regimes for which the present algorithm works compared to the previous work. There does not seem to be a clean way to state how the present work improves over the previous work. There's a discussion in the paper (paragraph starting at Line 209, and more in the full paper) mentioning situations where the algorithm of the present paper provides much better guarantees than the past work (these are formulated in terms of Dobrushin condition). I am unable to independently evaluate how much of an advance this constitutes over the past work because I don't know how significant this regime is for the present context of RBMs. Is the choice of complexity parameters in Definition 2 dictated by the proof techniques? Do they have any other significance? The results on supervised RBMs are not particularly good in experiments, presumably because the distributional assumption are not satisfied by the data.

Correctness: The proofs appear to be correct though I didn't check all details.

Clarity: Generally well-written.

Relation to Prior Work: Well done.

Reproducibility: Yes

Additional Feedback: Line 191: (since general RBMs can represent arbitrary distributions) I don't see why this is true. The RBM as defined below line 41 is clearly restricted in what distributions it can represent. "We also note that the parameters of the RBM are not identifiable even given 58 an infinite number of samples, so our goal for learning the RBM is generally speaking to learn the 59 distribution or related structural properties (e.g. the Markov blankets of the nodes in X)." Please give a reference or prove it.

[Author Response · NeurIPS 2020]

**Author Response: From Boltzmann Machines to Neural Networks and Back Again** We thank the reviewers for their input. First, we answer a few high-level questions asked by the reviewers:

**Relation to practical algorithms for training RBMs** We view this as a first step in importing tools from graphical models to combine with neural net methods. These new tools should be useful to improve over classic heuristics like contrastive divergence training, which have sometimes disappointing performance in practice.

**Experimental results** The purpose of our experiment was to compare our method with others with similar complexity and limitations, like classical CD training of RBMs, to which this method compares reasonably well. For reasons of comparison, this model has some obvious handicaps, e.g. it only uses binary units so it has to view grey as a probability and thus cannot really understand textures, it has no convolutional structure, etc; this is a proof-of-concept experiment.

**Motivation for Supervised RBM Learning** As discussed in the related work section, the distributional assumptions in the literature under which we have good theoretical neural net learning results are unfortunately very narrow. Many results either depend poorly on key parameters, or rely very strongly on Gaussianity of the input which is unlikely to hold in practice. In comparison, assuming data comes from some natural family of graphical models may be a more reasonable assumption, since a lot of data in practice is clearly structured and energy-based methods have seen a lot of success in modern image processing and machine learning. Our approach can be viewed as theoretically understanding how learning about the input distribution could play a role in supervised learning tasks.

**Context for Theorem 4 (Structure Learning of RBMs)** There is a huge literature on structure learning in the context of graphical models with no latent variables (e.g. references 3-8 and many more). However, for latent variable models like RBMs there is much less theory. For RBMs, the main previous works here are the cited results of Bresler, Koehler, and Moitra and the work of Goel; these results both require ferromagneticity (non-negative interactions). By viewing the distribution on observed variables as an MRF, it is possible to use general MRF-learning results (as in [5] and [8]), but the runtime of these methods is fairly poor: it is $n^{O(d)}$ where $d$ is the max degree of a hidden unit.

Reviewer 4 asked for more context as to when the $\ell_1$-norm is smaller than the degree. Specifically in the context of RBMs, Hinton's guide [1] says on page 9 "Care should be taken to ensure that the initial weight values do not allow typical visible vectors to drive the hidden unit probabilities very close to 1 or 0 as this significantly slows the learning" and suggests very small edge weights for initialization – standard deviation $0.01$ in his example. In the context of a $d$-sparse RBM, as $d \to \infty$ we need the edge weights to scale down if we want the typical input to a hidden unit to be size $O(1)$ (e.g. if the visible units behave like they are independent, we need the edges to scale like $1/\sqrt{d}$). So the $\ell_1$ norm will be much smaller than the degree.

**Why are the complexity parameters in Definition 2 natural?** $\ell_1$-norm is perhaps the most popular complexity measure in the literature on learning graphical models (see e.g. [8]) because $\ell_1$ regularization encourages sparsity, and sparse graphical models allow for drawing more powerful inferences (about conditional independence, etc.) than dense ones do. Mathematically, it's also the most natural because the spins $X$ and $H$ live in the $\ell_\infty$ unit ball and $\ell_1$ is the dual norm. Finally, $\ell_1$-norm bounds are the main assumption studied in the sampling literature (see lines 209-220 and Remark 4).

Next, we answer the remaining technical questions asked by the reviewers:

- Reviewer 4 asked for justifications of the following statements about RBMs: 1) they can represent arbitrary distributions (line 191), and 2) their parameters are not identifiable (i.e. impossible to estimate even with an infinite amount of data). Note that these are both statements about the observed distribution $X$, since $H$ is unobserved; i.e. the joint distribution of $(X, H)$ is not arbitrary (it's an Ising model), however the marginal on $X$ is arbitrary as long as we have enough hidden units. In prior work, [18] showed that any order $r$ Markov Random Field can be represented as the distribution on the observed units of an RBM with hidden units of degree $r$. By the Hammersley-Clifford Theorem, every distribution $p(x)$ on $\{\pm 1\}^n$ with $p(x) \neq 0$ for all $x$ is an order $r$ MRF for some $r \leq n$, so such a distribution can be exactly represented as the marginal over observable units in the RBM. They also gave several examples of RBMs which have different parameters but represent the same distribution (e.g. by having hidden units which cancel out each others effects on the observed units), which proves non-identifiability.

- Reviewer 1 asked for references for the statement on line 57, that learning full-observed Ising models is well understood. Some recent references with results for learning Ising models under weak assumptions are [5-8].

- Answers to the other questions asked by reviewer 1: on lines 209-220, here $d$ stands for the maximum degree of nodes in the RBM. In Theorem 2, $\ell$ is indeed the logistic loss. In Examples 1 and 2, $\eta$ indeed represents the maximum $\eta$ such that all 2-hop neighbors are $\eta$-nondegenerate.

- Reviewer 3 questions: On line 129, the feature map is all monomials of degree up to $D$, $d$ is a typo. Line 100: $n_V$ here is the number of visible units, $n_H$ is the number of hidden units.

[Meta-Review · NeurIPS 2020]

The initial scores in the four reviews were all in favour of accepting, although not strongly. The paper studies a relevant problem, presenting a new algorithm with performance guarantees and almost matching lower bounds. However some questions were raised regarding, for example, connections to other work and practical algorithms, and also more technical issues. The authors provided a detailed reply. After discussion among the reviewers, their concerns were partially answered, leading to somewhat stronger support for accepting.